**Title page**
**Soil fauna: key to new carbon models**
**Authors**
**Juliane Filser[1*], Jack H. Faber[2], Alexei V. Tiunov[3], Lijbert Brussaard[4], Jan Frouz[5],**
**Gerlinde De Deyn[4], Alexei V. Uvarov[3], Matty P. Berg[6], Patrick Lavelle[7], Michel Loreau[8],**
**Diana H. Wall[9], Pascal Querner[10], Herman Eijsackers[11], Juan José Jiménez[12]**
[1]Center for Environmental Research and Sustainable Technology, University of Bremen, General and Theoretical
Ecology, Leobener Str. – UFT, D-28359 Bremen, Germany.
email: filser@uni-bremen.de
* Corresponding author
[2]Wageningen Environmental Research (Alterra), P.O. Box 47, 6700 AA Wageningen, The Netherlands
email: jack.faber@wur.nl
[3]Laboratory of Soil Zoology, Institute of Ecology & Evolution, Russian Academy of Sciences, Leninsky prospekt 33,
119071 Moscow, Russia
email: av.uvarov@hotmail.com
email: a_tiunov@mail.ru
[4]Dept. of Soil Quality, Wageningen University, P.O. Box 47, 6700 AA Wageningen, The Netherlands
email: lijbert.brussaard@wur.nl
email: gerlinde.dedeyn@wur.nl
[5]Institute for Environmental Studies, Charles University in Prague, Faculty of Science, Benátská 2, 128 43 Praha 2,
Czech Republic
email: jan.frouz@natur.cuni.cz
[6]Vrije Universiteit Amsterdam, Department of Ecological Science, De Boelelaan 1085, 1081 HV Amsterdam, The
Netherlands
email: m.p.berg@vu.nl
[7]Université Pierre et Marie Curie, Centre IRD Ile de France, 32, rue H. Varagnat, 93143 Bondy Cedex, France
email: patrick.Lavelle@ird.fr
[8]Centre for Biodiversity Theory and Modelling, Station d'Ecologie Théorique et Expérimentale, UMR 5321 CNRS &
Université Paul Sabatier, 2, route du CNRS, 09200 Moulis, France
email: michel.loreau@ecoex-moulis.cnrs.fr
[9]School of Global Environmental Sustainability & Dept. Biology, Colorado State University, Fort Collins, CO
80523-1036, USA
email: Diana.Wall@ColoState.EDU
[10]University of Natural Resources and Life Sciences, Department of Integrated Biology and Biodiversity Research,
Institute of Zoology, Gregor-Mendel-Straße 33, A-1180 Vienna - Austria
email: pascal.querner@boku.ac.at
[11]Wageningen University and Research Centre, PO Box 9101, 6700 HB Wageningen, The Netherlands
email: Herman.Eijsackers@wur.nl
[12] ARAID, Soil Ecology Unit, Department of Biodiversity Conservation and Ecosystem Restoration, IPE-CSIC,
Avda. Llano de la Victoria s/n, Jaca 22700 (Huesca), Spain
email: jjimenez@ipe.csic.es



## Abstract

Soil organic matter (SOM) is key to maintaining soil fertility, mitigating climate change, combatting land degradation, and conserving above- and below-ground biodiversity and associated soil processes and ecosystem services. In order to derive management options for maintaining these essential services provided by soils, policy makers depend on robust, predictive models identifying key drivers of SOM dynamics. Existing SOM models and suggested guidelines for future SOM modelling are defined mostly in terms of plant residue quality and input and microbial decomposition, overlooking the significant regulation provided by soil fauna. The fauna control almost any aspect of organic matter turnover, foremost by regulating the activity and functional composition of soil microorganisms and their physical-chemical connectivity with soil organic matter. We demonstrate a very strong impact of soil animals on carbon turnover, increasing or decreasing it by several dozen percent, sometimes even turning C sinks into C sources or vice versa. This is demonstrated not only for earthworms and other larger invertebrates but also for smaller fauna such as Collembola. We suggest that inclusion of soil animal activities (plant residue consumption and bioturbation altering the formation, depth, hydraulic properties and physical heterogeneity of soils) can fundamentally affect the predictive outcome of SOM models. Understanding direct and indirect impacts of soil fauna on nutrient availability, carbon sequestration, greenhouse gas emissions and plant growth is key to the understanding of SOM dynamics in the context of global carbon cycling models. We argue that explicit consideration of soil fauna is essential to make realistic modelling predictions on SOM dynamics and to detect expected non-linear responses of SOM dynamics to global change. We present a decision framework, to be further developed through the activities of KEYSOM, a European COST action, for when mechanistic SOM models include soil fauna. The research activities of KEYSOM, such as field experiments and literature reviews, together with dialogue between empiricists and modellers, will inform how this is to be done.

## Keywords

Carbon turnover, soil organic matter modelling, soil invertebrates, aggregate formation, spatial heterogeneity, regional differences, ecosystem engineers, COST Action KEYSOM

## 1. Introduction

Despite continuous refinement over the past decades, estimates of the global carbon cycle still show large discrepancies between potential and observed carbon fluxes (Ballantyne et al., 2012; Schmitz et al., 2014). Soils contain more carbon than the atmosphere and aboveground vegetation together (Brevik et al., 2015) and play an important role for many of the recently adopted UN Sustainable Development Goals (Keestra et al., 2016). Therefore soil organic matter (SOM) modelling is key to understanding and predicting changes in global carbon cycling and soil fertility in a changing environment. SOM models can facilitate a better understanding of the factors that underlie the regulation of carbon cycling and the persistence of SOM. The predictive power of current global SOM models is, however, limited, as the majority relies on a relatively restricted set of input parameters such as climate, land use, vegetation, pedological characteristics and microbial biomass (Davidson and Janssens, 2006). Other parameters, such as the leaching of organic matter or soil erosion of organic matter have been suggested for improving model predictions, and recent research has demonstrated what drastic effects e.g. living roots (Lindén et al., 2014) and soil fungi (Clemmensen et al., 2013) exert on SOM persistence. In an overview on the performance of SOM models, none of 11 tested models could predict global soil carbon

accurately, nor were 26 regional models able to assess gross primary productivity across the US
and Canada (Luo et al., 2015).
Some years ago Schmidt et al. (2011) proposed eight "key insights" to enrich model predictions
on the persistence of SOM. However, they ignored a major component of SOM dynamics, soil
fauna, which play a fundamental role in most of the insights they propose (e.g. Fox et al., 2006;
Jimenez et al., 2006; Osler and Sommerkorn, 2007; De Deyn et al., 2008; Wilkinson et al., 2009).
By moving through and reworking soil, feeding on living plant roots, detritus and all types of
microorganisms growing on these, soil animals are intimately involved in every step of SOM
turnover. Omission of soil fauna from SOM models will, therefore, hamper the potential
predictive power of these models.
In a review focusing mostly on large mammals, terrestrial herbivores and aquatic ecosystems,
Schmitz et al. (2014) recently called for "animating the carbon cycle". Bardgett et al. (2013)
argued that differential responses of various trophic groups of aboveground and belowground
organisms to global change can result in a decoupling of plant-soil interactions, with potentially
irreversible consequences for carbon cycling. A correlative large scale field study has suggested
that including soil animal activities could help clarify discrepancies in existing carbon models (de
Vries et al., 2013). Similar attempts to connect animal activity to carbon cycling occurred in the
past (e.g. Lavelle and Martin, 1992; Lavelle et al., 1998; Lavelle and Spain, 2006; Osler and
Sommerkorn, 2007; Brussaard et al., 2007; Sanders et al., 2014), without any further change in
the structure of carbon models. This was partly due to a lack of communication between
modellers and experimenters, but also because the magnitude of animal effects on SOM
dynamics remains poorly quantified (Schmitz et al., 2014).
Here we use the 'key insights' proposed by Schmidt et al. (2011) as a basis to review current
evidence and to identify research needs on the relationship of soil fauna to SOM dynamics. Our
review justifies the relevance of incorporating the soil fauna into SOM models. How important
animal activities are for manifold geological and pedological processes has been reviewed
repeatedly (e.g. Swift et al., 1979; Wilkinson et al., 2009), but carbon turnover – which is highly
dynamic and both directly and indirectly affected by animals – never had been the focus. Due to
their prime role in most processes in soil (Briones, 2014) we mostly focus on earthworms, but
also give examples for other groups of soil fauna whose role in C turnover appears to be much
more relevant than thought thus far (e.g. David, 2014). We point out regional differences in
climate, soils and land use with respect to soil fauna composition, abundance and activity and
derive implications for SOM modelling. Finally, we introduce a new COST Action (ES 1406)
that is working on the implementation of soil fauna into SOM models, also exploring the pros and
caveats in such a process.

## 2. Key insights

The eight 'key insights' compiled by Schmidt et al. (2011) are shown in Fig. 1, together with the
most important activities of soil animals affecting them. As many animal-mediated processes are
tightly interconnected, they also matter for most of these insights. For instance, aggregate
formation in faeces simultaneously affects molecular structure, humic substances, physical
heterogeneity and soil microorganisms. In the following text we briefly summarize the role of
animal activities for each of the 'key insights'. As a more detailed example of animal impacts on
SOM turnover, we consider their role on soil aggregate formation in a separate section.

### 2.1. Molecular structure

The molecular structure of root exudates and dead organic matter is modified during
metabolisation, decomposition and associated food web transfer, both by microorganisms and
soil fauna. Prominent examples are the release of ammonium by bacterivorous protozoans and

nematodes, due to their higher C:N ratio compared to their bacterial prey (Osler and Sommerkorn, 2007), or the strong mediation of the direction and rate of humus formation by soil animals (see 2.2). Recently, the significant impact of eight different species of ants over 25 years on mineral dissolution and accumulation of calcium carbonate has even been discussed in the context of geoengineering and carbon sequestration (Dorn, 2014).

Many soil animals ingest and process SOM (and accompanying microorganisms) in their gut system, where it is partly assimilated with the help of mutualistic gut microflora and partly egested. Metabolisation alters the chemical structure of ingested SOM (Jiménez and Lal, 2006; Hedde et al., 2005; Coulis et al., 2009; Frouz, 2015b; Schmitz et al., 2014) and, consequently, the decomposition dynamics of animal faeces, which can be a substantial component of SOM (humus). Humification as such renders SOM less decomposable (Blume et al., 2009; Dickinson, 2012) whereas the alkaline milieu in invertebrate midguts accelerates mineralisation (e.g. Li and Brune, 2007).

For instance, earthworm casts have species-specific NIR spectral signatures, indicating presence of specific organic compounds (Hedde et al., 2005). Under grass/legume pasture they are characterized by significant enrichment of slightly-altered plant residues in the sand particle size (> 53 $\mu$m). CPMAS 13C NMR (Cross-Polarization Magic Angle Spinning Carbon-13 Nuclear Magnetic Resonance) spectra showed that earthworm casts and surrounding soil were dominated by carbohydrates, with a decrease of O-alkyl C and an increase of alkyl C with decreasing particle size (Guggenberger et al., 1996). Moreover, earthworms likely possess a unique capability of neutralizing plant polyphenols that otherwise strongly decrease decomposition rates of fresh plant litter (Liebeke et al., 2015). Micro- and mesofauna excrete ammonium or dissolved organic carbon (Filser, 2002; Fox et al., 2006; Osler and Sommerkorn, 2007), and affect the quantity of microbial metabolites (Bonkowski et al., 2009). Gut passage, defecation, excretion together with bioturbation by macro- and mesofauna facilitate humification and decomposition, altering also nutrient stoichiometry (Bohlen et al., 2004). These modifications in the molecular structure of SOM due to soil fauna activity have significant effects on its dynamics (Swift et al, 1979; Guggenberger et al., 1995; Blume et al., 2009; Dickinson, 2012 and other references related to points 1 and 2 in Table 1).

## 2.2. Humic substances

As stated above, humification increases SOM stability. The term "humic substances" (here defined as very large and highly complex, poorly degradable organic molecules with manifold aromatic rings; Lehmann and Kleber, 2015) may be considered problematic by part of the scientific community: neither is the concept itself clear, nor is there any evidence that the often mentioned highly complex large organic molecules play any relevant role in organic matter stabilisation under natural conditions (Schmidt et al., 2011; Lehmann and Kleber, 2015). However, here we stick to it when referring to the "insights", simply for reasons of consistency with the article our argumentation is based on Schmidt et al. (2011). We acknowledge that "humus" or "humic substances" represent a continuum of more or less decomposed dead organic matter of which energy content and molecule size mostly should decrease over time, and that water solubility, sorption to the mineral matrix and accessibility for microorganisms are highly relevant for OM turnover (Lehmann and Kleber, 2015).

Humic substances are formed during the gut passage: organic matter in young soils and humic horizons almost completely consists of soil animal faeces (Lavelle, 1988; Martin and Marinissen, 1993; Brussaard and Juma, 1996). Humus forms mainly comprise animal casts, e.g. casts of ants, isopods, millipedes, beetle larvae or termites in deadwood, of insect larvae and spiders in leaf litter, or casts of collembolans, mites and enchytraeids in raw humus. In his review, David (2014) considered macroarthropod casts being a factor of partial SOM stabilisation, rather than hotspots of microbial activity. The dark colour of casts (compared to the ingested organic material) visually demonstrates the strong chemical OM modification in animal guts, which is

accompanied by a substantial physical modification. Clay-humus complexes, physically
protecting organic matter (Jiménez and Lal, 2006), are mainly faeces of earthworms and
diplopods (see also Sect. 2.5 on physical heterogeneity). Due to differences in feeding
preferences, gut microflora, SOM quantities consumed etc. of soil animals, their faeces vary in
size, shape and quality not only between fauna groups, but also between species within one group
(see Sect. 3 on aggregate formation). Discovering the important role of animal faeces in
humification is essential to improve our understanding of carbon dynamics in soil.

## 2.3. Fire-derived carbon

Fire-derived carbon is chemically highly condensed and thus often hardly degradable. However,
its stability in soil is variable and still poorly understood (Schmidt et al., 2011; Wang et al.,
2016). Two of the factors identified by a meta-analysis on the stability of biochar in soil were
association with aggregates and translocation in the soil profile (Wang et al., 2016), which are
both strongly affected by soil fauna (see Sect. 2.5, 2.6 and 3). Microbial recolonisation of burned
sites is mediated by wind and soil animals that survived in soil or emigrated from neighbouring
areas, e.g. by macro- and mesofauna, birds and mice (Malmström, 2012; Zaitsev et al., 2014).
Besides, soil fauna also ingest the charcoal particles (Eckmeier et al., 2007; see Table 1). Due to
animal activity, charcoal is sorted by size and translocated down the soil profile. Mice and
earthworms (Eckmeier et al., 2007) and the tropical earthworm *Pontoscolex corethurus*
(Topoliantz and Ponge 2003; Topoliantz et al., 2006) had been suggested as responsible for rapid
incorporation of charcoal into the soil. Quantitative data are however scarce (Table 1). In spite of
potentially great importance, the effect of soil animals on the fate of the "black carbon" in soil
remains practically unknown (Ameloot et al., 2013).

## 2.4. Roots

Roots not only represent a major input pathway of carbon into soil, but together with associated
microflora they also have a large influence on the turnover dynamics of existing soil carbon
(Finzi et al., 2015). Roots preferably grow in existing soil cavities (Jiménez and Lal, 2006),
mostly formed by soil fauna (Wilkinson et al, 2009). Both burrowing and non-burrowing soil
animals have a strong impact on root growth, allocation, length and density (Brown et al., 1994;
Bonkowski et al., 2009; Arnone and Zaller, 2014). Animal grazing of root bacteria and
mycorrhiza affects their activity and community composition, and animal excreta are enriched in
micronutrients and selectively affect plant nutrition (Brown, 1995; Filser, 2002; Brussaard et al.,
2007). Root herbivores and rhizosphere grazers affect C allocation of roots (Wardle et al., 2004)
and largely regulate nutrient acquisition and plant productivity (Bonkowski et al., 2009). Not
only root herbivores, but also saprotrophic/microbivorous soil animals may obtain a significant
proportion of energy from plant roots (Pollierer et al., 2007). This suggests an animal-mediated
regulatory loop that connects plant roots and SOM.

## 2.5. Physical heterogeneity

Schmidt et. al. (2011) considered the physical disconnection between decomposers and organic
matter to be one reason for SOM persistence in deep soil. Yet, physical heterogeneity in soils
occurs at all spatial scales, and animals play a fundamental role in the distribution of organic
matter and associated microorganisms. According to body size, decomposers act at various
spatial scales, from micro-aggregates to landscapes (Ettema & Wardle, 2002; Jouquet et al.,
2006). They fragment organic residues, perform bioturbation, distribute dead organic matter and
generate smaller and larger organic matter hot spots (e.g. faecal pellets, ant and termite mounds).
Mounds and burrows made by soil fauna are obvious signs of physical heterogeneity created by
ecosystem engineers (Meysmann et al., 2006; Wilkinson et al., 2009; Sanders et al., 2014), which
significantly affect microorganisms, plants (Chauvel et al., 1999; Frelich et al., 2006), aggregate
stability (Bossuyt et al., 2005; 2006), hydraulic properties (Bottinelli et al., 2015; Andriuzzi et al.,
2016), sorption and degradation of sparingly soluble organic compounds (Edwards et al., 1992;
Bolduan and Zehe, 2006) and C emissions (Wu et al., 2013; Lopes de Gerenyu et al, 2015).
Earthworms in particular feed on organic and mineral parts of the soil and mix them (Eckmeier et
al., 2007; Wilkinson et al., 2009). The resulting clay-organic matter complexes considerably
increase SOM retention of soils (Jiménez and Lal, 2006; Fox et al., 2006; Brussaard et al., 2007),
although C loss from fresh casts is much higher than from surrounding soil (Zangerlé et al.,
2014). The impact on soil processes and physical heterogeneity varies considerably between
different groups of ecosystem engineers (Jouquet et al., 2006; Bottinelli et al., 2015). For
instance, some earthworm species strongly affect their physical environment while others are
more linked to the soil organic matter content (Jiménez et al. 2012).

### 2.6. Soil depth

In most soil types, pore volume, carbon content, associated biotic processes and temperature
variability strongly decrease with depth whereas other parameters such as bulk density and water
content increase − all of which significantly affect SOM turnover rates. The depth of organic
horizons varies with soil type, from almost zero to several metres. Thus, Schmidt et al. (2011)
identified soil depth as another "key insight". Yet, digging animals play a key role in the
development of soil depth. A considerable part of physical heterogeneity are animal burrows that
can reach several meters deep. Bioturbation (e.g. by earthworms, termites, ants, beetle and
Diptera larvae, spiders, solitary bees and wasps, snails, isopods and amphipods, puffins, lizards,
porcupines, pigs, moles, voles, rabbits, foxes, or badgers) is a key process to the formation of soil
depth, soil structure and associated C translocation, as shown by several examples in Table 1 and
reviewed e.g. by Wilkinson et al. (2009).

### 2.7. Permafrost

In permafrost soil up to $1,672 * 10^{15}$ g carbon is stored (Tarnocai et al. 2009). Organism activity
is mostly restricted to the short periods of time when the upper cm of the soil isare thawed. Due
to unfavourable environmental conditions (resulting in low animal biomass, activity and
diversity), there is only little impact of fauna in permafrost soils (De Deyn et al., 2008). However,
fauna invasions, especially of the above-mentioned soil engineers, due to soil melting in tundra
and boreal forests are likely to have drastic effects (Frelich et al., 2006; Van Geffen et al., 2011).
Data on earthworm invasions in North American forests (Bohlen et al., 2004; Frelich et al., 2006;
Eisenhauer et al., 2007) show that they must be taken into consideration in carbon-rich soils,
particularly in melting permafrost soils (Frelich et al., 2006; Schmidt et al., 2011) where they
may affect many soil functions.

### 2.8. Soil microorganisms

After roots, microorganisms constitute by far the largest share ofbiomass in soil biota.
Accordingly, they have a crucial role in SOM turnover. They consume root exudates, dead
organic matter, attack plants and animals as pathogens or support them as mutualists. Finally,
microorganisms are the most important food source for the majority of soil animals, and to a
considerable part also for aboveground insects and vertebrates. Soil fauna comprise ecosystem
engineers as well as an armada of mobile actors connecting elements of the soil system,
mediating microbial processes (Briones, 2014). Countless isopods, ants, termites, enchytraeids,
microarthropods, nematodes or protozoans make large contributions to SOM turnover
underground (Persson, 1989; Filser, 2002; Wardle et al., 2004; Fox et al., 2006; Osler and
Sommerkorn, 2007; Wilkinson et al., 2009; Wu et al., 2013). They affect the activity and
community composition of soil microorganisms in multiple ways such as feeding, burrowing,
facilitating the coexistence of different fungal species (Crowther et al., 2011) or by modifying
micro-habitat conditions. Litter comminution by detritivores increases SOM accessibility for
microorganisms, and propagules are dispersed with body surface and casts. The gut environment
provides protected microsites with modified biotic and abiotic conditions, which increase
bacterial abundance substantially – e.g. by three orders of magnitude in earthworm guts (Edwards
and Fletcher, 1988). Grazing affects microbial biomass, activity and community composition, and
animal excreta modify nutrient availability for microorganisms (Brown, 1995; Filser, 2002).
Table 1 contains quantitative examples of animal activity taken from different biomes and land-
use types, showing that earthworms alone strongly affect each of the 'key insights'. However,
much smaller soil animals can also have substantial effects (Table 1). It has to be kept in mind
that the separation of animals' effects according to the insights is somewhat arbitrary as the
associated soil processes are often interconnected. This is particularly obvious for molecular
structure, humic substances, roots, physical heterogeneity, soil depth and microorganisms:
metabolisation implies by definition an alteration of the molecular structure, often associated with
the formation of humic substances. The stability of the latter has a very strong association with
physical protection, and whether metabolisation of dead organic matter occurs at all depends on
its horizontal and vertical distribution. For instance, earthworms will (a) translocate dead organic
matter both vertically and horizontally, (b) transform part of it via metabolisation, (c) mix
ingested OM with minerals, thus affecting its physical protection, (d) increase and alter the
microbial community and (e) affect hydraulic properties and aeration of the soil through digging
and tunnelling, which has an immediate impact on the activity of microorganisms and on root
growth.
As this example illustrated only the most important aspects of interacting processes,, the next
section provides a more elaborate overview on aggregate formation.

## 3. Aggregate formation

The modern view on the stability of organic matter in soils requires a thorough understanding of
aggregate structure and formation including the role of soil biota (Lehmann and Kleber, 2015).
Soil aggregation is the process by which aggregates of different sizes are joined and held together
by different organic and inorganic materials. Thus, it includes the processes of formation and
stabilisation that occur more or less continuously, and can act at the same time. With clay
flocculation being a pre-requisite for soil aggregation, the formation of aggregates mainly occurs
as a result of physical forces, while their stabilisation results from a number of factors, depending
in particular on the quantity and quality of inorganic and organic stabilising agents (Amézketa,
323    1999).
By bioturbation, feeding and dispersal of microbial propagules soil animals regulate all of the
above forces and agents, and are therefore a crucial factor in the formation and stabilisation of
soil aggregates. Earthworms, many insect larvae and other larger fauna may stabilise aggregate
structure by ingesting soil and mixing it intimately with humified organic materials in their guts,
and egesting it as casts or pellets (Tisdall and Oades, 1982; Oades, 1993).
Earthworms have a direct and fast impact on microaggregate formation and the stabilisation of
new C within these microaggregates (Bossuyt et al., 2005) (Table 1). There are several
mechanisms to explain the increase of micro- and macroaggregate stability by earthworms, but no
mechanism has been quantified in relation to population size yet. Effects are related to ecological
groups of earthworms, associated with feeding habit, microhabitat in the soil profile, and burrow
morphology. However, irrespective of this classification, species may enhance or mitigate soil

compaction (Blanchart et al., 1997; Guéi et al., 2012). The tensile strength of casts (roughly defined as the force required to crush dried aggregates, i.e. an indirect measure of physical SOM protection) appears to be species dependent: for example, the casts of *Dendrobaena octaedra* have a lower tensile strength compared to those of *L. terrestris* (Flegel et al., 1998). Similarly, organic carbon and water-stable aggregation was significantly higher in casts of *L. terrestris* than in casts of *A. caliginosa* (Schrader and Zhang, 1997).

Some research, however, suggests that earthworm activity can also evoke soil degradation. Shipitalo and Protz (1988) proposed that ingestion of soil by earthworms results in disruption of some existing bonds within micro-aggregates and realignment of clay domains. Therefore, fresh casts are more dispersible than uningested soil, contributing to soil erosion and crusting. Significant improvement in the water stability of fresh, moist casts only occurs when incorporated organic debris from the food sources is present and when moist casts are aged or dried. Nevertheless, in the long term, casting activity enhances soil aggregate stability.

However, our understanding of the contribution of soil fauna to aggregate formation and stabilisation is limited, and mostly qualitative in nature. Different methodologies complicate the comparison among aggregate stability data (Amézketa, 1999). Data in terms of functional response to density are limited as many studies have been conducted in arable systems, where the diversity and abundance of soil animals are reduced as a consequence of tillage, mineral fertilizers and pesticide use. Recently, some studies have emerged. A negative correlation between earthworm abundance and total macroaggregates and microaggregates within macroaggregates in arable treatments without organic amendments could be linked to the presence of high numbers of *Nematogenia lacuum*, an endogeic species that feeds on excrements of other larger epigeic worms and produces small excrements (Ayuke et al., 2011). Under the conditions studied, differences in earthworm abundance, biomass and diversity were more important drivers of management-induced changes in aggregate stability and soil C and N pools than differences in termite populations. Another study highlighted that in fields converted to no-tillage earthworms incorporated C recently fixed by plants and moved C from soil fragments and plant residues to soil aggregates of >1 mm (Arai et al., 2013). Thus, soil management practices altering fauna activities may have a significant effect on the re-distribution of soil organic matter in water-stable aggregates, impacting agronomically favourable size fractions of water-stable macro-aggregates, and water-stable micro-aggregates which are the most important source of carbon sequestration (Šimanský and Kováčik, 2014).

## 4. Regional differences in climate, soils and land use

In a global meta-analysis spanning several continents, García-Palacios et al. (2013) show that across biomes and scales the presence of soil fauna contributes on average 27% to litter decomposition. Depending on the situation this contribution can be substantially lower or higher. For instance, the authors report an average increase in decomposition rates of 47% in humid grasslands whereas in coniferous forests this figure amounts to only 13%. The high impact of soil fauna in humid grasslands is all the more important as such grasslands are among those ecosystems that are most severely affected by global environmental change (Chmura et al., 2003; Davidson and Janssen, 2006).

Many of our examples refer to earthworms and temperate regions as they have been studied most intensively. However, we suggest that any dominant group of soil fauna, irrespective of body size or the ability to create larger soil structures, may substantially affect carbon dynamics. Table 1 gives a number of respective case studies. The key players and specific effects of soil animals vary across space (Fig. 2), with increasing importance for SOM dynamics in humid-warm and nutrient-limited conditions (Persson, 1989; Filser, 2002; Wardle et al., 2004; Fox et al., 2006; Osler and Sommerkorn, 2007; De Deyn et al., 2008; Briones, 2014). Once key players in a given ecosystem have been identified as relevant for being included in SOM models (see Sect. 6 and

Fig. 3), more detailed information on their biology is required, in particular on their activity, their ecological niche and corresponding tolerance limits. All this varies with species, and often extremely within one systematic group. Variation in drought or soil temperature towards limiting conditions will first increase (stress response, e.g. downward migration) and then strongly decrease activity (mortality or transition to inactive resting stage). Some key players will exhibit high abundance and be extremely active throughout the year (Wilkinson et al., 2009), others might only be moderately relevant during a short period of time; the contribution of a third group might be considered insignificant.

Also ecosystem engineers differ between soil types, biomes and land-use types, from rodents and ants in dry areas to termites, earthworms and millipedes in tropical rainforests. They consume different types of organic matter, make deep or shallow, narrow or wide burrows, and differ in aggregation behaviour (e.g. more or less regularly distributed earthworms versus distinct ant nests and termite mounds). Accordingly, their role in SOM re-distribution and turnover differs as well. In cold ecosystems – where, together with wetlands and peatlands, the majority of terrestrial carbon is stored (Davidson and Janssens, 2006) – the response of detritivores to climatic change is expected to be most pronounced (Blankinship et al., 2011). Melting of permafrost soil might lead to northward expansion of soil macro-invertebrates, associated with accelerated decomposition rates (van Geffen et al., 2011). Further examples are shown in Table 1.

More information is needed on how existing abiotic and biotic constraints to SOM decomposition will vary with changing climate and in different regions (Davidson and Janssens, 2006). Finally, human activity comes into play: any significant land use change, particularly soil sealing and conversion of native forest to agricultural land, has dramatic consequences for abundances and species composition of soil communities. The same holds true for management intensity and pollution (Filser et al., 1995; Filser et al., 2002; Filser et al., 2008; De Vries et al., 2012). Yet, even seemingly harmless activities can be significant, as we will show for the case of fishing in the end of Sect. 5 – pointing out the relevance of human activities for soil fauna beyond impact on global warming and land use change. How we address all this biogeographical and ecological variation is shown in Sect. 5 and 6.

## 5. Implications for modelling

As there is no unambiguous scientific support for the widespread belief in "humic substances", the question how long organic carbon remains in soil is largely related to a) physical protection and b) how often the once photosynthesized dead organic matter is recycled in the soil food web. For both processes soil animals are of great importance, as we have shown above. Biomass and abundance of soil animals are generally constrained by temperature, humidity and food (living or dead organic matter). However, the effects of these constraints on their activity are not simply additive, nor is there any simple relation between biomass and activity. For example, despite overall unfavourable conditions for the majority of soil organisms, burrowing activity in deserts can be extremely high (Filser and Prasse, 2008). Moreover there is increasing evidence that fauna effects on energy and nutrient flow can be at least partly decoupled from other abiotic and biotic factors (Frouz et al., 2013). De Vries et al. (2013) even concluded that "Soil food web properties strongly and consistently predicted processes of C and N cycling across land use systems and geographic locations, and they were a better predictor of these processes than land use". This implies that knowledge of fauna may increase our prediction power. The thermodynamic viewpoint makes the issue even more relevant: reaction speed increases with temperature, but most soil organisms are rather adapted to relatively cool conditions and might thus be pushed beyond their niche limits – with eventually negative consequences on their activity, see Sect. 4. Changes in climate (Blankinship et al., 2011), land use (Filser et al. 2002; Tsiafouli et al., 2014), resource availability and biotic interactions (De Vries et al., 2012; see Table 2) alter the distribution, community composition, activity and associated impact of soil animals on distribution and turnover rate of SOM (Wall et al., 2008) to the extent that underlying

assumptions of SOM models may no longer be valid (Swift et al., 1998; Bardgett et al., 2013;
Schmitz et al., 2014). Therefore it is opportune to include approaches that have been developed
during the past decades (Filser, 2002; Jiménez and Lal, 2006; Osler and Sommerkorn, 2007;
Brussaard et al., 2007; Meysmann et al., 2006; Wall et al., 2008; Sanders et al., 2014). For
instance, Lavelle et al. (2004) implemented earthworm activity in the CENTURY model. For this
purpose, observations on long-term incubated earthworm casts and sieved control had been used
as a reference. Afterwards earthworm activity was simulated with CENTURY by replacing the
active and slow soil C decomposition rates of the model with those obtained by calibration with
the control soil. The simulations revealed a 10% loss of the slow C pool within 35 years
compared to the original model without earthworms.
Without considering the role of animals, models are less accurate: in a field study spanning four
countries from Sweden to Greece, soil food web properties were equally important as abiotic
factors and predicted C and N cycling processes better than patterns of land use (De Vries et al.,
2013). In their study, earthworms enhanced $CO_2$ production whereas Collembola and
bacterivorous nematodes increased leaching of dissolved organic carbon. Mechanistic
experiments confirm that earthworms have a detrimental effect on the greenhouse gas balance
under nitrogen-rich conditions (Lubbers et al., 2013) and under no-till (Lubbers et al., 2015).
Inclusion of group-specific diversity of mesofauna in models of global-scale decomposition rates
increased explained variance from 70 to 77% over abiotic factors alone (Wall et al., 2008). Also
García-Palacios et al (2013) provide additional evidence on the argument that soil fauna activity
is not merely a product of climate, soil properties and land use but an independent parameter.
These examples indicate that the actors that play an important role in SOM dynamics should be
considered in SOM models.
Model parameters are often measured *in situ* at relatively large spatial scales – at least compared
with the size or activity range of most soil animals. As a result, the fauna effect is *de facto*
included, although not appreciated (Swift et al., 1998). However, in many cases parameters are
measured or extrapolated by combining *in situ* methods (e.g. monitoring of gas flux or litterbag
experiments) and *ex situ* techniques such as laboratory experiments at controlled, highly
simplified conditions. Especially the results of the latter may be sensitive to neglecting soil fauna.
A relationship between animal activity and C turnover may vary with scale, for instance when
soil properties or animal abundance differ at larger distance. However, as data are often
insufficient, it will be context-dependent if the inclusion of fauna is sensible or not (see Sect. 6).
On the other hand, not taking explicitly into account the spatial heterogeneity created by soil
fauna in field measurements might lead to substantial errors in calculating carbon budgets (Wu et
al., 2013; Lopes de Gerenyu et al, 2015). It is thus crucial to develop sound (and biome-specific)
strategies for combining *in-* and *ex-situ* measurements as parameters in more realistic SOM
models.
Next to space, scale effects also apply to temporal patterns – which poses a great challenge for
SOM modelling as most studies refer to rather short periods of time. We illustrate this by the
comparatively well studied impact of invasive earthworms. The meta-analysis of Lubbers et al.
(2013) suggests that the effect of earthworms on total SOC contents is on average relatively
small. In contrast, in certain situations earthworms can strongly affect greenhouse gas emission.
These data were however mainly obtained in relatively short-term experiments. Over a period of
months to years and even decades, earthworms can reduce C decomposition by physical
protection of C in ageing casts (Six et al., 2004, see Table 1).
Thus, underlined long-lasting effects of invasive earthworms on the total SOC storage cannot be determined
with certainty in short-term experiments, whereas field observations are rather controversial. For
instance, Wironen and Moore (2006) reported ca. 30% increase in the total soil C storage in the
earthworms-invaded sites of an old-growth beech-maple forest in Quebec. Other studies (e.g.
(Sackett et al., 2013; Resner et al., 2014) suggest a decrease in C storage. Zhang et al. (2013)
introduced the sequestration quotient concept to predict the overall effect of earthworms on the C
balance in soils differing in fertility, but the question remains strongly understudied.

These well documented examples of the impact of earthworms on soil C storage are related to invasive species. The presence of these species cannot be inferred directly from the climatic, soil and vegetation properties. The distributions of European invasive earthworms in North America, North European forests or South Africa are largely driven by human activity. Often fishing (due to lost baits), imported plants or potting material of colonizing farmers (Reinecke, 1983) are more important for these than habitat transformation – without human's help earthworms are not active invaders (Stoscheck et al., 2012; Tiunov et al., 2006; Wironen and Moore, 2006). Thus the presence of earthworms can be an environment-independent parameter of SOM dynamics.

Another fundamental issue in the large-scale approach is often neglected: When including the effects of the soil fauna implicitly, this assumes that the soil fauna will always have the same effects under the same conditions, and hence that the soil fauna are essentially static. This assumption is increasingly unrealistic in a fast-changing world where both biodiversity and the climate are changing at accelerated paces, and where we are likely to witness major reorganisations of plant, animal and microbial communities. Therefore explicit representation of the soil fauna, where possible, should increase the predictive ability of SOM models.

Given the fact that this issue had been raised decades back (see above) it appears somewhat astonishing that attempts to pursue it have not yet made any significant progress. We believe there are mainly three reasons for this: a) missing information, b) too much detail, irrespective of spatial scale, and c) too little communication between empiricists and modellers. This is why we decided to bring into life a COST Action as an appropriate instrument to bridge these gaps. The next section gives an overview on it.

## 6. Ways to proceed: COST Action ES 1406

Based on the arguments compiled here, a COST Action entitled "Soil fauna - Key to Soil Organic Matter Dynamics and Modelling (KEYSOM)" was launched in March 2015 ([http://www.cost.eu/COST_Actions/essem/ES1406](http://www.cost.eu/COST_Actions/essem/ES1406)). An interdisciplinary consortium of soil biologists and biogeochemists, experimenters and modellers from 23 European countries plus the Russian Federation and the USA cooperates to implement soil fauna in improved SOM models as a basis for sustainable soil management. The main aim of KEYSOM is to test the hypothesis that the inclusion of soil fauna activities into SOM models will result in a better mechanistic understanding of SOM turnover and in more precise process descriptions and output predictions of soil processes, at least locally. A number of workshops address key challenges in experimentation and modelling of SOM and soil fauna and support research exchange and access to experimental data. Special attention is given to education of young scientists. The Action comprises four Working Groups (WG) with the following topics:

1. Knowledge gap analysis of SOM – soil fauna interactions;
2. Potentials and limitations for inclusion of soil fauna effects in SOM modelling;
3. Data assemblage and data sharing;
4. Knowledge management and advocacy training.

After an intensive and enthusiastic workshop held in Osijek, Croatia in October 2015, first activities included compilation of literature, the setup and permanent update of a website ([http://keysom.eu/](http://keysom.eu/)). Meanwhile short-term scientific missions for early-career scientists have been launched ([http://keysom.eu/stsm/KEYSOM-STSMs-are-open-for-application](http://keysom.eu/stsm/KEYSOM-STSMs-are-open-for-application)), aiming for complementing the Action's activities. The second workshop was held in Prague in April 2016.

Next to a first compilation of knowledge gaps in this article, present activities of KEYSOM involve

- a literature review on biome-specific effects of soil fauna impact on SOM turnover

-     a literature review on the impact of soil fauna other than earthworms on SOM turnover
-     a compilation of the potentials and limitations of existing SOM models
-     the development of a simple SOM model that also explicitly incorporates soil animals and
associated processes in it, based on the current state of knowledge exchange between
empiricists and modellers within KEYSOM
-     the preparation of a common European-wide field study into the impact of soil fauna
composition and abundance on SOM breakdown, distribution and aggregate formation,
which will start in autumn 2016
-     the preparation of a summer school, to be held in early October 2016 in Coimbra,
542                Portugal

Fig. 3 illustrates the present state of our interdisciplinary discussions, providing a roadmap how
SOM models could be supplemented with the effects of soil fauna. In the first phase, empiricists
(Fig. 3A) and modellers (Fig. 3B) work in parallel. Mutual exchange between these groups is
guaranteed by the regular workshop meetings such as in Osijek and Prague.
The stepwise approach functions like a decision tree, with various feedback loops and options at
every step if and how known effects of soil fauna could be implemented into SOM. It also
identifies under which circumstances additional research (literature review or experimental
studies) needs to be initiated before proceeding further. As many existing models, also the new
model should have a modular structure so that different modules can be used and combined
according to the respective underline{biome- and scale-specific scenario} (Fig. 3C). It can also be seen that
underline{we do not aim to include every detail everywhere}: in some situations (Fig. 3A) the impact of soil
fauna on SOM dynamics might be too small (or existing information too scanty) to be included,
and not all input parameters will be feasible or relevant at each scale (miniature in Fig. 3C). This
keeps the model manageable, and also flexible enough to allow for underline{more precise predictions in}
underline{critical scenarios,} like in the case of earthworm invasions sketched in Sect. 5. We generally think
that focusing on such critical scenarios (analogous to e.g. global biodiversity hotspots) is a crucial
precondition for underline{well-informed management decisions, one of the final aims of KEYSOM.}
As an example, box no. 1 in Fig. 3A stands for the first literature review in the above list.
Depending on the outcome, for each biome a decision will be made if the impact of fauna on
SOM turnover is unknown, relevant or low. In the first case, more research is needed, in the last
case the faunal effect can be ignored. Depending on the outcome of additional research, the
knowledge base will be improved and the decision between ignoring and proceeding further can
be made anew. If a strong effect is expected, the next question (box no. 2 in Fig. 3A) will be
addressed and so forth.
Once the procedure in Fig. 3A has reached box no. 4, intensive exchange with modellers (Fig.
3B) is mandatory to identify the relevant model parameters and the type of functional relationship
(box 5). Mechanistic aspects (such as chemical transformation in the gut, physical protection
within aggregates or impact on hydraulic soil properties via digging) are of prime importance
here as each of these examples may have different effects on C turnover. Effects of fauna
abundance or biomass (in comparison to presence-absence) on the underline{shape of the function} will be
addressed as well. Note, however, that to date necessary data for such an approach appear to be
limited (García-Palacios et al., 2013). – In the meantime, the modellers will have developed a
basic model structure and compare it with the structures of existing SOM models concerning
potentials and limitations for including fauna effects (Fig. 3B).
The second phase (Fig. 3C) starts with the practical tests of the collected model parameters
(boxes 6 and 7), using data that have been compiled by then by WG 3, allowing for selecting the
best model (box 8). At this point, spatial scale comes into play, which is likely to be the most

critical issue: As we have seen also while preparing this article, existing data on the impact of soil fauna on SOM turnover are highly diverse, from short-term and often highly artificial experiments at controlled conditions to large-scale correlative field studies in all kinds of different environments (and with a strong bias what comes to certain biomes). The type of relationship between faunal abundance and SOM turnover will in most cases vary with scale. If data for different scales are not available (box 9), further research is needed. In the second case, one can proceed with boxes 10 and 11.

Importantly, the idea is not to include the fauna in every situation everywhere. Rather we aim at identifying critical hotspots and scenarios (see above) where faunal activities play a crucial role in SOM turnover, as demonstrated in Sect. 5. Due to the abovementioned differences between biomes and scale effects, these scenarios will be biome- and scale-specific. An example is shown in the lower left corner of Fig. 3C. For Biome A, hydraulic properties have been identified to be crucial for SOM dynamics. Thus, data are needed on animals that affect these, such as digging earthworms or rodents. Instead, the analyses for Biome B have revealed aggregate structure and microorganisms being most relevant – claiming for respective data at the small scale. On a larger scale such data for microorganisms might not be available, which implies proceeding with aggregate structure alone.

Overall, the whole approach requires a modular model structure, allowing for using different models according to the respective situation and data availability. This is what WG 2 is currently developing. – Certainly all the research outlined here cannot be done within one single COST Action. Based on the outcome of our work, we hope to come up with a more detailed roadmap how to further proceed to improve SOM modelling. This roadmap, together with what could be achieved with the limited resources of KEYSOM, will provide information material, decision tools and management options for decision makers and politicians (WG 4).

## 7. Conclusions and outlook

Understanding and modelling SOM is essential for managing the greenhouse gas balance of the soil, for land restoration from desertification, for sustaining food production and for the conservation of above- and belowground biodiversity and associated ecosystem services (Nielsen et al., 2015). Soil animal abundance, biodiversity, species traits and interactions are crucial for SOM turnover (Chauvel et al., 1999; Bohlen et al., 2004; Wardle et al., 2004; Wall et al., 2008; Uvarov, 2009). In Table 2 we give recommendations how the known impact of soil fauna on SOM turnover could be used for improving carbon models. Due to the pronounced differences with respect to climate, soil and land use outlined above, it is important that these recommendations are considered region- and scale-specific, taking into account the key players and their specific activities in the respective area.

## Author contribution

J. Filser wrote the article, prepared Fig. 1 and 3 and the tables and compiled the contributions from all co-authors. These are listed according to their quantitative and qualitative impact on the manuscript, except for J.J. Jiménez who was placed last as he is the chair of COST Action ES 1406 (KEYSOM). L. Brussaard suggested including Fig. 2.


## Acknowledgements


Three anonymous referees are acknowledged for their critical comments which significantly
contributed to the revision of the original manuscript. We thank Antje Mathews for compiling the
references and editing the manuscript. Many thanks to Karin Nitsch for linguistic proof-reading.
Oxford University Press and Wiley and Sons are acknowledged for the permission to include Fig.
2. This paper is a contribution to the COST Action ES1406 (KEYSOM) lead by the first and last
author. A lot of the writing was inspired by the lively discussions within the workshop meetings
of KEYSOM – thanks to all contributors! We thank COST Association for financially supporting
collaboration and networking activities across Europe.

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

**Tables**
**Table 1.** Quantitative examples of the impact of earthworms and selected groups of other soil fauna on
soil properties and processes involved in soil organic matter (SOM) turnover. If not mentioned otherwise,
any numbers or percentages refer to the control without fauna. Selected particularly striking examples are
printed in bold.

| Insight* | Examples | Source |
|---|---|---|
| | Earthworms | |
| 1. Molecular structure | An indicator of lignin degradation in earthworm casts was twice that of the surrounding soil | Guggenberger et al., 1995 |
| 2. Humic substances | Introduced earthworms can double microaggregate formation and the stabilisation of new C in the topsoil | Marashi and Scullion, 2003; Six et al., 2004 |
| | C protection is promoted by microaggregates within large macroaggregates, and earthworms can add 22% anew to this C pool | Bossuyt et al., 2005 |
| | Exclusion of earthworms reduced SOC accumulation by 0 (at 0-10 cm depth) to 75% (at 30-40 cm depth), associated with a decrease in percentage of water-stable aggregates | Albrecht et al., 2004, cited in Schmidt et al., 2011 |
| | In organic layers of a Canadian aspen forest, in locations with earthworms, N (1.5–0.8%) and especially C concentrations (25.3–9.8%) were strongly reduced, together with C/N ratio (16.7–13.2) and soil pH (6.5–6.1); in brackets: control values vs. values with earthworms. **This suggests a shift towards a faster cycling system, resulting in a net loss of C from the soil and turning Northern temperate forests from C sinks into C sources** | Eisenhauer et al., 2007 |
| 3. Fire-derived carbon | Small charcoal particles from burned plots after one year increased by 21% in 0-1cm depth. One year later they were concentrated in earthworm casts at the soil surface, after 6.5 years such casts were found at 8 cm depth | Eckmeier et al., 2007 |
| 4. Roots | **Presence of earthworms in a continuous maize plot in Peruvian Amazonia increased the organic C input from roots by 50%** | Jiménez et al., 2006 |
| 5. Physical heterogeneity | Up to 50% of soil aggregates in the surface layer of temperate pastures are earthworm casts | Van de Westeringh, 1972 |
| (see also insights no. 2, 3, 6 and 7) | Mull-type forest soil top layers and wooded savanna soils consist almost entirely of earthworm casts | Kubiena, 1953; Lavelle, 1978 |
| | **Earthworm inoculation in pastures on young polder soils completely removed within 8-10 years the organic surface layer, incorporated it into deeper layers, creating an A horizon. This affected manifold measures, increasing e.g. grass yield by 10%, root content in 0-15% from 0.38 to 1.31 g dm$^{-3}$, C content in 0-20 cm from 1.78 to 16.9 kg C * $10^3$ ha$^{-1}$, and water infiltration capacity from 0.039 to 4.6 m 24 h$^{-1}$. In turn, penetration resistance at 15 cm depth decreased from 35 to 22 kg cm$^{-2}$.** | Hoogerkamp et al., 1983 |
| | In average temperature pasture and grasslands, earthworms cast 40-50 t ha$^{-1}$ year$^{-1}$ on the surface and even more below surface | Lee, 1985 |
| | **Passage of a tropical soil through the gut of the invading earthworm *Pontoscolex corethrurus* reduced macroporosity from 21.7 to 1.6 cm$^3$ g$^{-1}$, which exceeded the effect of mechanically compacting the same soil at $10^3$ kPa (resulting macroporosity: 3 cm$^3$g$^{-1}$)** | Wilkinson et al., 2009 |
| | After invasion of European earthowrms into a Canadian aspen forest a thick layer of their cast material (thickness up to 4 cm) on top of organic layers was developed | Eisenhauer et al., 2007 |
| 6. Soil depth | Burrows of anecic earthworms are up to several meters deep and last for many years | Edwards and Bohlen, 1996 |

\* According to Schmidt et al. (2011)

**Table 1.** (continued)

| Insight* | Examples | Source |
|---|---|---|
| Earthworms | | |
| 7. Permafrost and boreal areas | **Earthworm invasions in boreal forests completely transformed mor to mull soils and significantly altered the entire plant community** | Frelich et al., 2006 |
| 8. Soil microorganisms | Earthworms may lower actual microbial activity (by 11-23%) but markedly (by 13-19%) optimize microbial resource utilization. | Scheu et al., 2002 |
| Ants and termites | | |
| 2. Humic substances | **In a degraded marsh in NE China, ant mounds were $CH_4$ sinks, contrary to the control soils which were $CH_4$ sources (-0.39 – -0.19 mg vs. 0.13 – 0.76 m$^{-2}$ h$^{-1}$)** | Wu et al., 2013 |
| 5. Physical heterogeneity | Ant and termite mounds can occupy up to 25% of the land surface | Bottinelli et al., 2015 |
| 5. Physical he-terogeneity and 6. soil depth | **Underground nests of leafcutter ants (e.g. *Atta* spp.) can cover up to 250 m² and extend down to 8 m., which is associated with a massive impact on forest vegetation** | Correa et al., 2010 |
| Collembola | | |
| 8. Soil microorganisms | Grazing by Collembola affected community composition of ectomycorrhizal fungi and on average reduced $^{14}CO_2$ efflux from their mycelia by 14% | Kanters et al., 2015 |
| | Grazing by *Protaphorura armata* at natural densities on AM fungi disrupted carbon flow from plants to mycorrhiza and its surrounding soil by 32% | Johnson et al., 2005 |
| | **The presence of a single Collembola species may enhance microbial biomass by 56%** | Filser, 2002 |
| | At elevated temperature, litter decay rates were up to 30% higher due to Collembola grazing | A'Bear et al., 2012 |
| Various or mixed groups | | |
| 1. Molecular structure | Microbial grazing by Collembola or enchytraeids alone enhanced leaching of $NH_4^+$ or DOC by up to 20%[5] | Filser, 2002 |
| | **Feeding by millipedes and snails reduced the content of condensed tannins in three Mediterranean litter species from 9–188 mg g$^{-1}$ dry matter to almost zero** | Coulis et al., 2009 |
| | Long-term mineralisation of fauna faeces may be slower than the mineralisation of litter from which the faeces were produced. This decrease in decomposition rate corresponds to a decrease in the C:N ratio and in the content of soluble phenols. | Frouz et al., 2015a,b |
| | Due to stoichiometric constraints, soil animals tend to reduce the C concentration of SOM, but increase N and P availability. About 1.5% of the total N and P in the ingested soil was mineralized during gut passage in humivorous larvae of the scarabaeid beetle *Pachnoda ephippiata*. In *Cubitermes ugandensis* termites, the ammonia content of the nest material was about 300-fold higher than that of the parent soil. | Li et al., 2006; Li and Brune, 2007; Ji and Brune, 2006 |
| 2. Humic substances | In a laboratory experiment, activity of earthworms, Collembola, enchytraeids and nematodes in coarse sand liberated >40% from the insoluble C pool as compared to the control | Fox et al., 2006 |
| | Radiolabelled proteins and phenolic compounds in litter are faster transformed to humic acids (as revealed by alkaline extraction and acid precipitation) via feces of Bibionidae (Diptera) than from litter not eaten by fauna | Frouz et al., 2011 |
| | The quantitative contribution of invertebrates (mainly beetles and termites) to wood decomposition ranges between 10-20% | Ulyshen, 2014 |
| | **Depending on fungal and animal species (Collembola, isopods and nematodes), grazing on fungi colonising wood blocks altered (mostly decreased) their decay rates by more than 100%. Isopods and nematodes had opposite effects in this study**. | Crowther et al., 2011 |

* According to Schmidt et al. (2011)

| Insight* | Examples | Source |
|---|---|---|
| | Various or mixed groups | |
| 2. Humic substances (continued) | Carbon and nitrogen losses from soil followed by drought and rewetting were substantially affected by microarthropod richness, which explained 42% of the residual variance. | De Vries et al., 2012 |
| 5. Physical heterogeneity | **Bioturbation rates of soil animal groups typically range between 1 and 5 Mg ha$^{-1}$ y$^{-1}$ but may reach up to 10 (crayfish, termites), 20 (vertebrates), 50 (earthworms) and > 100 Mg ha$^{-1}$ y$^{-1}$ (earthworms in some tropical sites), which is equivalent to maximum rates of tectonic uplift** | Wilkinson et al., 2009 |
| 8. Soil microorganisms | In the course of a 2.5-yr succession, fauna activities (especially of nematodes and mesofauna during the first year, and later of earthworms) accelerated microbial decomposition of clover remains in an arable soil by 43% | Uvarov, 1987 |
| | Depending on vegetation, animal group and climate, soil animals directly or indirectly increased C mineralisation between 1% and 32%. However, intensive grazing by fungal feeders may even reduce C mineralisation | Persson, 1989 |

* According to Schmidt et al. (2011)

**Table 2.** "Insights" (compiled after Schmidt et al., 2011) for future soil organic matter models and
recommendations for further improvements by implementing effects of soil fauna

| SOM modelling element ("Insight") | Recommendations* |
|---|---|
| 1. Molecular structure | Incorporate the knowledge on the structure of organic substances and element concentrations in faunal casts and excreta in SOM decay rate models. Consider linkage between C and N cycling mediated by fauna. See 8. |
| 2. Humic substances | Add physical and chemical stability of casts, patterns of their microbial colonisation and degradation dynamics. See 1, 3, 5, 6, 7, 8. |
| 3. Fire-derived carbon | Include recolonisation and inoculation potential of surviving soil fauna and adjacent fauna. Initiate studies on the impact of fauna on the fate of black carbon (fragmentation, gut, casts, decomposition, and recolonisation). |
| 4. Roots | Add activity of bioturbators, rhizosphere microbial grazers and root herbivores. See 1, 5, 6, 8. |
| 5. Physical heterogeneity | Consider spatial and physicochemical heterogeneity created by soil fauna, including consequences of soil aggregation and dis-aggregation (e.g. bulk density, infiltration rate, preferential flow, casts). See 1, 2, 6, 8. |
| 6. Soil depth | Incorporate burrowing depth and annual transport rates of bioturbators and animal-induced spatial heterogeneity of old and young carbon in the deep soil. See 5. |
| 7. Permafrost | For warming scenarios, take into account short- and long-term invasion effects, particularly of earthworms and enchytraeids. |
| 8. Soil microorganisms | Add microbial grazer effects, effects on microorganisms during gut passage and faunal impact on C and N coupling. See 1-7. |

* Recommendations refer to site-specific keystone groups of animals (dominating in terms of biomass or impact; see
Fig. 2). Their prevalence is determined by climate, bedrock and land use (e.g. rodents or ants in deserts, earthworms in
temperate grasslands or microarthropods and enchytraeids in acidic Northern forests).

**Figure Captions**

**Figure 1.** Main animal-mediated processes (boxes) affecting the eight insights (symbols)
identified by Schmidt et al. (2011) that should be considered for improving SOM models

**Figure 2.** Dominant soil types and characteristic soil forming invertebrates across biomes (major
global change threats are shown in italics). MAT = mean annual temperature, MAP = mean
annual precipitation. Sources for data and biomes see Brussaard et al. (2012).
© John Wiley and Sons. Reprint (slightly modified) by kind permission from John Wiley and
Sons and Oxford University Press.

**Figure 3.** Flow scheme for an improved understanding of the role of soil fauna for soil organic
matter (SOM) turnover. This scheme is basically followed within the COST Action ES 1406
(KEYSOM). Activities in A) and B) run parallel, followed by C) which ends with an improved
SOM model. Exemplarily shown are scenarios for two biomes. Further explanations see text.


**Figures**

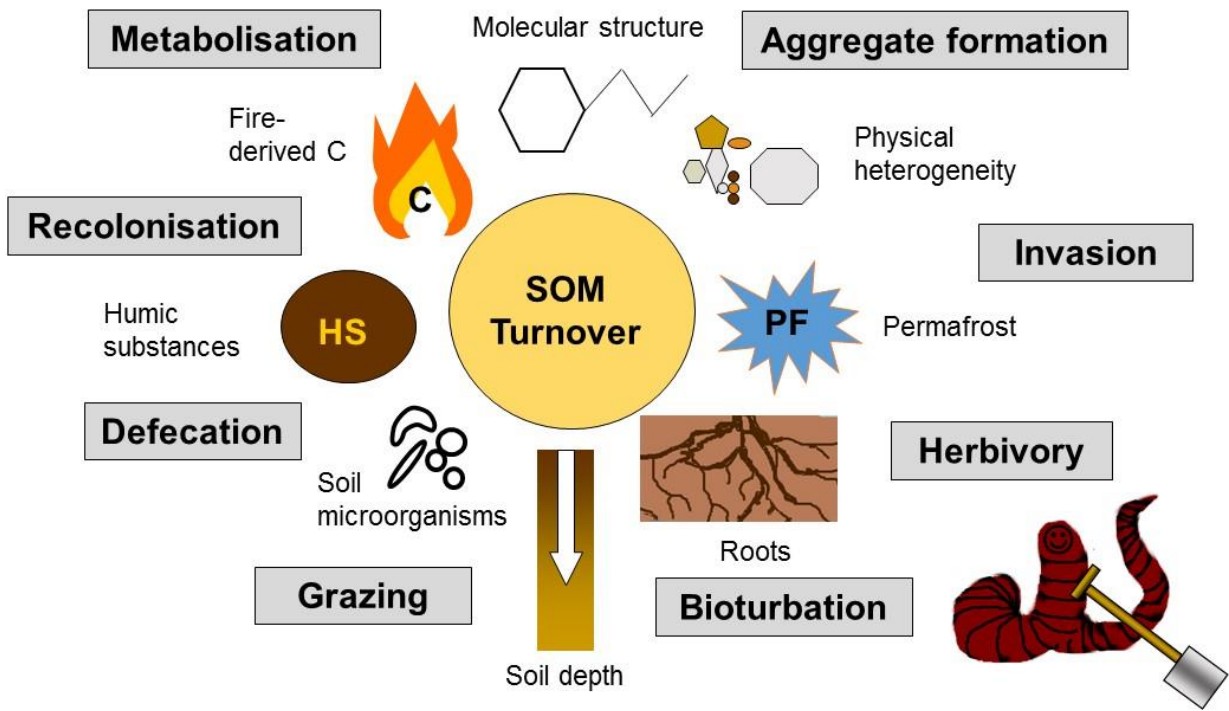

**Figure 1**





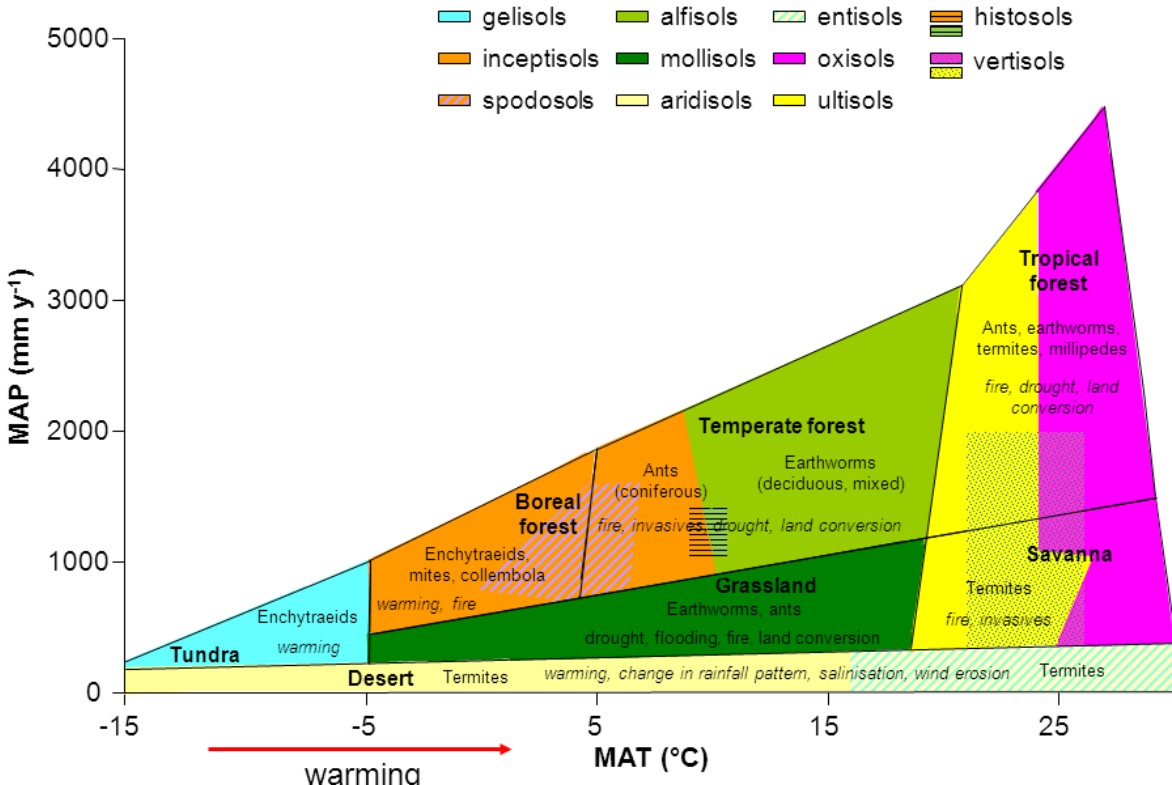


**Figure 2**


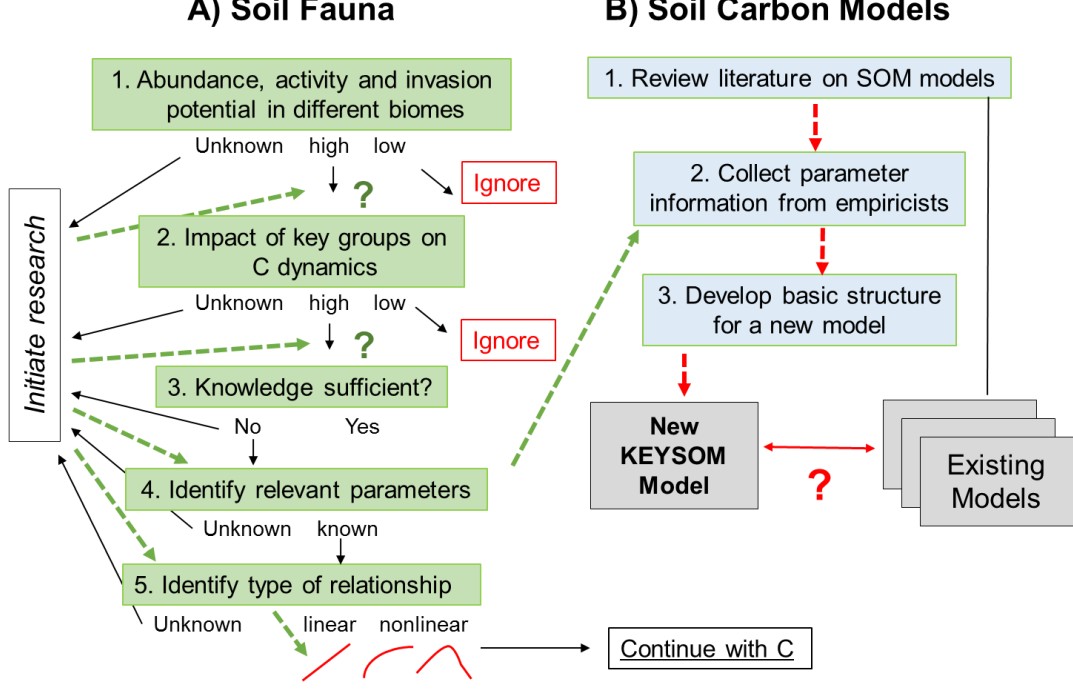


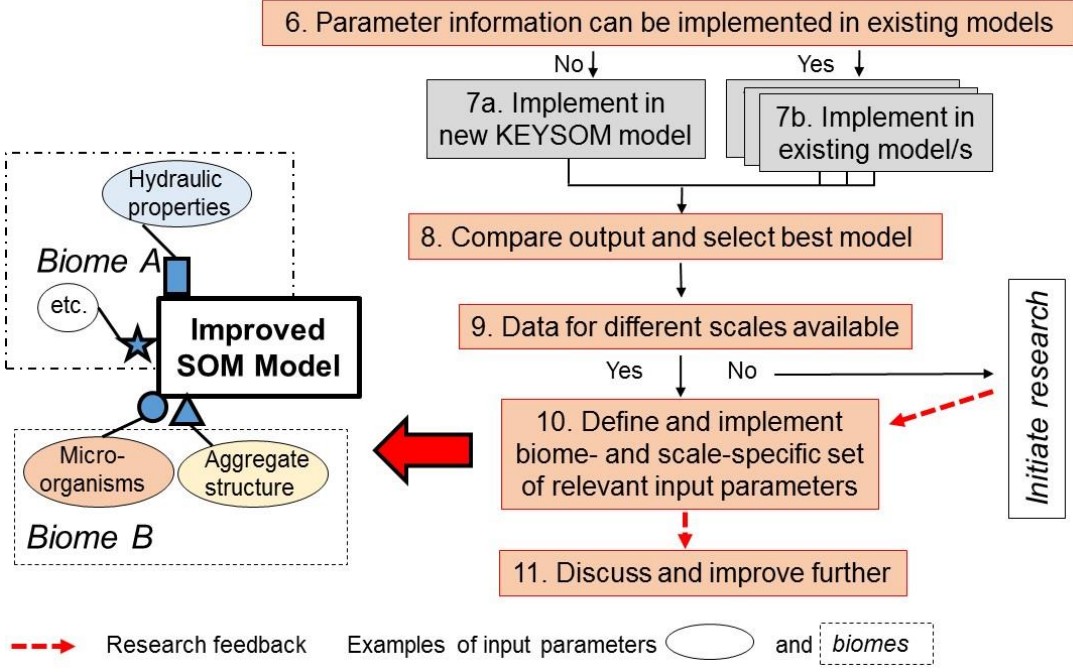



**Figure 3**
