# Peer review of "Title page Soil fauna: key to new carbon models Authors"

_SOIL, 2016_

## Referee Comment (RC1) · Anonymous Referee #1 · 10 May 2016

This MS makes the important argument that understanding the exchanges and fates of carbon between soils and ecosystems requires deliberate consideration of soil animals and their actions in determining the build-up of soil organic matter. Carbon modeling has been dismissive of animal effects largely because of an under-appreciation of the multiplier (knock-on) effects that animals can have on geophysical processes. The MS provides an almost encyclopedic, detailed review of the myriad, mechanisms by which soil animals influence the amount and fate of organic matter by altering geophysical processes in terrestrial soils. This clearly-written MS covers the relevant literature and thereby provides the reader a useful entre into the scope of work on soil organismal biology as it relates to soil carbon.

The MS is intended to convince carbon modelers that they need to consider the effects of soil fauna in order to enhance the predictive power of the models and thereby develop more accurate accounting of soil carbon. But, here is where I think the MS

falls short. While it presents important ideas, the material isn't communicated in a way that can be appreciated by modelers, who routinely tend not to be empiricists, let alone experimentalists, and therefore likely wouldn't concern themselves with the fascinating, but minutiae of, detailed mechanisms presented in the MS (or wouldn't know where to begin to incorporate the details into models). The MS instead would likely resonate most with soil biologists who are fascinated by the details, and thus the MS tends to "preach to the choir" so to speak. The MS would be strengthened if it played more directly to the perspectives or needs of modelers. But, here is where the authors need to decide on which direction to go.

From where I stand, making a convincing case for including effects of soil animals in modeling needs to provide one of two important pieces of information (if not both).

The first would be to provide modelers with a clearer sense of how discrepant their model predictions are because they don't include animal effects. That is, how much does the presence vs. absence of groups of soil animals influence the amount of soil carbon that is stored or lost. The MS gets at it a bit when discussing earthworm effects (line 344-346). But, there is scant other evidence provided to support the argument (beginning on line 347) that without considering the role of animals models will be less accurate. It would be helpful to know quantitatively how inaccurate the models will be if animal effects are not included (i.e., how much of a difference in carbon balance estimates is there?). I appreciate that this may be tough to do because empiricists and experimentalists aren't accustomed to examine soil biology and relate it to quantitative estimates of carbon balance in a form that is useable by modelers. (There is a lesson for empiricists here too).

Second, as the MS correctly points out, many of the biological details presented in the MS are ignored or simply subsumed as "parameters" in models that describe big-scale processes. This is typically done as a matter of mathematical convenience because abstracting a complex process as a parameter keeps the model reasonably tractable. But, accounting for animal effects in ways described in the MS requires characterizing

those processes in terms of model functions, functions in which the levels of soil carbon due to a specific mechanism (e.g. soil bioturbation, or aggregate formation) vary with the abundance of the animals performing the mechanism. However, converting parameters into new functions can be a daunting exercise from two standpoints. First, empiricists and experimentalists tend to examine processes in terms of effects due to presence/absence of species and often do not vary animal abundance to measure what form a function should take. Second, empiricists are enamored by biological details, but often don't give priority to which details might matter more than others. This can cause concern to modelers because including each and every detail can make the models vastly complex, therefore making model output extremely difficult to validate and therefore understand. So, given heterogeneity in soil properties across large geographic spaces, do we need to know accurately variation in local soil molecular structure, or local root processes, or local physical heterogeneity or local aggregate formation to inform regional carbon budgets? If yes, then how? What I am driving at is that the MS would be strengthened if it provided a better road map of what processes should be an immediate priority to include in modeling and what level of detail needs to be included in the models. This road map could be strengthened if the MS could offer a sense of what the functional forms of the processes might look like (i.e., can we assume linearity? Must we consider nonlinearity? If nonlinearity, then what form should the nonlinear function take?). Most importantly, if accounting for spatial variation in animal effects matters, then what is a reasonable spatial scale over which one can assume that animal effects are reasonably similar. That is, it would be impossible for models of regional carbon budgets to account for heterogeneity on a m^2 x m2 basis. What spatial scale could be reasonable: km^2 x km^2, 10 km^2 x 10 km^2? Solving this scaling problem is perhaps the most critical issue when trying to align models with empiricism. In my experience, this is what causes the biggest rift between modeling and empiricism: empiricists again tend to focus on details of very fine spatial heterogeneity and disagree with efforts to subsume that heterogeneity in to a reasonable large-scale spatial approximation.

Ultimately, the issues raised in the MS are not merely issues that should be of concern just to modelers. Empiricists need to appreciate the challenges and demands of modeling and provide empirical input that can help meet the challenge by tailoring empirical estimates and analyses to explicitly inform modeling. There was a large movement afoot in ecology in the 1990's to do a better job of melding modeling and empirical work. Modern ecology seems not to have heeded that too much. Perhaps the important message of this MS is that we need to begin heeding this a lot more going forward.

---

## Referee Comment (RC2) · Anonymous Referee #2 · 10 May 2016

The given manuscript (ms) is devoted to reviewing current scientific data on the relationships between soil fauna and soil organic matter (SOM) dynamics in the light of improving existing SOM models. By including into the models direct and indirect effects of soil fauna (for example such as feeding on plant residues and bioturbation of soil), it is thought by the authors to fundamentally enhance predictive outcomes of SOM models.

The ms has clear objectives and represents a good contribution to scientific progress in the interdisciplinary area of soil biologists/ecologists and modellers. It is considered a substantial number of relevant references, making the review very educational for students of any level and scientists.

At the current level of understanding soil biology processes, we do not need to justify the relevance of incorporating of soil fauna (or soil microorganisms, soil vertebrates etc.) into SOM models. Now the matter is to my knowledge how to implement these

data in theoretical models. Why these incorporations were not done before: due to lack of communication between modellers and biologists? Or it was not done due to lack of technical support (software) in modelling? Or biologists were not "convincing" enough for theoretical modellers?

Another matter in the ms is the introduction of the COST (KEYSOM) project? Is it like an advertisement of the project? What is the aim of this introduction? KEYSOM is a very interesting project, no doubts about that, but what is it for in the ms? Is it possible to justify the presence of such information in the review paper, please?

I like how the authors are trying to classify the effects of soil fauna on soil processes and properties into so called "key insights". It is nice to see that data are collected not only for earthworms effects but other soil invertebrates too, including Collembola, potworms, nematodes, ants, termites. Though, when looking at the source references in the Table 1 one gets an impression that there was not much done to quantitatively test effects of soil fauna in soil. It is the case, isn't it?

Figures and tables of the ms are important parts of the story. They inform reader in a graphical way very well and are self-explainable. Concerning figures, I would like to add that after printing they both look fuzzy (not sharp). In Fig.1 in permafrost "star" the abbreviation PF is not seen at all. In general this Fig has more a look that I would describe as a "whatever" look. Soil microorganism symbol looks like something else. Is it possible for you to make a picture where clearly, for example, with arrows or spatial restructuring, will be seen which "insight" affects which animal-mediated process? In Fig.2 unit of MAP is mm per year. What is this dot after mm?

Line 182. Needs to change references positions, year 2012 goes first. Line 254. Two times is used the word "on", please change the sentence. Line 290. Sentence "...diversity and abundance of soil animals IS reduced as....", should not be ARE here instead? Line 355. Needs to remove symbol % after 70. It should be only after 77%.

Overall the ms is written concisely, the language is fluent, precise, and grammatically

correct.

---

## Referee Comment (RC3) · Anonymous Referee #3 · 22 May 2016

GENERAL COMMENTS

This generally well written manuscript reviews the evidence for the need to incorporate soil fauna in SOM stabilization and dynamic models. It frames its review in the light of Schmidt et al.'s 2011 thought-provoking paper in Nature discussing the persistence of organic matter as an ecosystem property. As pointed out by other reviewers, the paper contains a wealth of biological detail. However, I think that this detail leads to the main message of the paper (which I assume is the need for the SOM modelling community to incorporate fauna) being lost.

To remedy this, I think the authors need to do four things:

a) If possible, clearly demonstrate how the presence or absence of fauna lead to changed model / empirical predictions of SOM dynamics. The controls for the examples in Table 1 are not always explained, so it is not clear how fauna are or are not

important. The modelling importance is touched upon around lines 347 but could be developed further, and explicit links could be provided in the earlier empirical review.

b) Clearly demonstrate that the already included model parameters (e.g. of climate, land use etc) do not adequately predict faunal composition (it is mentioned briefly with De Vries et al. 2013 on line 349). If land use/climate parameters can (generally) predict faunal composition then it is not immediately clear why fauna need including in SOM dynamic models if climate, land use etc can also predict SOM reasonably accurately [which I realise is not always the case]. As noted by another review, do the discrepancies in model results mean that fauna need including, or is it that other processes (e.g. leaching, particulate loss, litter inputs) could need refining to improve model accuracy instead.

Acting on this comment though does depend on model aims - if the aim of the model is to provide prediction, do they need to be mechanistically accurate (in the extreme, can they actually be statistical?). If however the model is aimed at mechanistic understanding, then the need for faunal incorporation potentially becomes clearer. As noted by the authors, this aspect may become even more important when trying to account for environmental change.

c) The authors mention in the Abstract that 'the contribution of soil fauna activities can be as high as 40 %' but it is not clear in the main text where this figure comes from. More importantly, what is the distribution of faunal importance to SOM dynamics/stability in ecosystems? Is it that there is one study demonstrating this level of importance, but others only show a negligible contribution? If so, then perhaps the importance of incorporation of faunal activity into SOM models is being overblown. [I personally think we do need to think about its incorporation, but the evidence presented here is not as clear as it could be] It may be that with the data currently available the distribution cannot be assessed. If so, then this aspect should, in my opinion, at least be discussed.

d) The title suggests 'new' models are required. It might be good if the authors could synthesize their review to put forward a conceptual model framework that contrasts with currently available frameworks e.g. RothC, CENTURY.

SPECIFIC COMMENTS

Abstract - line 60 - we suggest that inclusion of soil animal activities can fundamentally affect the predictive outcome of SOM models.

This is a very strong statement which I do not think has been clearly demonstrated in the paper at present. Perhaps addressing a) to c) above will help justify this. I can only see one example in the paper (referring to earthworms in CENTURY) and am not sure this is a 'fundamental' difference in prediction. What is a fundamental difference in prediction anyway - differing magnitude, differing direction (e.g. carbon source or sink), something else...?

Line 68 - I don't think there is any need to advertise the COST action in the Abstract (it occupies 6 lines out of 24), and it would be better to finish with a strong statement of what this review paper has found and its implications. I would also remove reference to this COST action at the end of the main text - it seems like a weak ending to the paper unless this is the main message you want to communicate.

MAIN TEXT

Line 104 - please provide more details on the correlative field study. Of what? What discrepancies?

Line 118 - point out regional differences in or of what? We already know about regional differences in climate, land use, soils?

Line 151 - "modifications in molecular structure have significant effects on its [SOM] dynamics." This is presumably the important point in terms of justifying incorporation for modellers yet there are no references backing this statement up. In relation to point c) above, what is the quantitative importance?

Line 155 - "the term humic substances is considered outdated"

This is a very strong statement yet many modellers continue to use these conceptual pools which reflect different rates of organic matter turnover. You could alienate readers with such phrasing, likewise on line 328 with "As there is no scientific support for widespread belief in humic substances".

Indeed, current research is utilizing mid infrared spectroscopy to examine particulate, humic, and resistant organic carbon, operationally defined pools which match well with conceptual pools in the models. These results have then been validated against observed changes in soil C following land use change (Paul et al. 2016 in review; see also Baldock et al 2013 Soil Research 51: 577-595). Does this means SOM models need a fundamental overhaul or does it depend on what you are trying to predict?

I don't think these statements are required for the broader message of the paper and would suggest tempering them.

Line 180 - what happens to fire derived carbon in absence of soil fauna?

Line 220 - what does discussion of soil depth mean for SOM dynamics. Make the links explicit here and in the other sections.

Line 241 - please put quantitative figures on 'a large contribution' i.e. define large.

Line 273 - why does tensile strength matter for SOM dynamics.

Line 289 - density of what?

Line 311 - how does Figure 2 demonstrate that specific effects of soil organisms differ across space? How has increasing importance in humid-warm and nutrient-limited conditions been demonstrated? Does the absolute / relative difference with and without soil organisms increase in these conditions? Or is it that soil organic matter dynamics are faster in humid warm conditions and so the presence of animals is confounded with these climatic conditions?

Line 344 - this sentence could be clearer - was the slow C pool maintained when earthworms were present in the model?

TECHNICAL CORRECTIONS

Line 393 - "A number of workshops" [not 'workshop'].

---

## Author Comment (AC1) · 1 Jul 2016

Dear editors,

please see our final comment in the text below and in the attached file.

Best wishes,

Juliane Filser, on behalf of all authors

We appreciate the high quality and detail of the referees' comments, in particular by referee #1, which have righteously criticized several points that we improved in the revised version. Looking at the wealth of information asked for, it should be clear that it is impossible to cover all this within a single review article – especially with one written mainly by empiricists and pointing out research needs. In the new version we have better pronounced why we think that these needs are still there although the issue as such had been raised decades back: these are mainly a) missing information, b)

[Figure]

too much detail, irrespective of spatial scale, and c) too little communication between empiricists and modellers.

Thus, and probably most importantly, we made now clear why we included the COST Action KEYSOM (which addresses right these issues and associated research needs), and added some more information on its contents and activities. Main aim of KEYSOM is to test the hypothesis that the inclusion of soil fauna activities into SOM models will result in a better mechanistic understanding of SOM turnover and in more precise process descriptions and predictions, at least locally. Current activities are mainly workshops and additional literature reviews, yet also modelling: Next to implementing fauna activity in already existing SOM models, a new basic SOM model structure is being developed. Ongoing literature reviews include faunal contribution to SOM turnover in different biomes and potentials and limitations of global SOM models for including soil fauna effects. Finally, a large-scale European field experiment into the impact of soil fauna composition and abundance on SOM breakdown, distribution and aggregate formation will start in autumn 2016.

Fig. 3 of the revised manuscript (see below) shows a form of a "roadmap" how KEYSOM is contributing to the challenges identified by our review. This is further explained in the accompanying text, where we also address a number of other issues raised by the referees. Please note that this roadmap mainly functions as a decision tree with potential actions and feedback loops at every step, for instance to initiate research (literature review or experimental studies) before proceeding further. Effects of fauna abundance or biomass (in comparison to presence-absence) on the shape of the function will be addressed as well (Fig. 3A, bottom). Note, however, that to date necessary data for such an approach appear to be limited (García-Palacios et al., 2013). As many existing models, also the new model should have a modular structure so that different modules can be used and combined according to the respective biome- and scale-specific scenario (Fig. 3C). It can also be seen that we do not aim to include every detail everywhere: in some situations (Fig. 3A) the impact of soil fauna on SOM

dynamics might be too small (or existing information too scanty) to be included, and not all input parameters will be feasible or relevant at each scale (miniature in Fig. 3C). This keeps the model manageable, and also flexible enough to allow for more precise predictions in critical scenarios, like in the case of earthworm invasions sketched in the next paragraph. We generally think that focusing on such critical scenarios (analogous to e.g. global biodiversity hotspots) is a crucial precondition for well-informed management decisions, one of the final aims of KEYSOM.

To further substantiate our point of view, we followed the suggestion to add more quantitative figures. This was already partly shown in the table or in the text of the MS and further supplemented. In addition we included a more elaborate example based on the comparatively well studied impact of invasive earthworms. The meta-analysis of (Lubbers et al., 2013, already cited in the MS) suggests that the effect of earthworms on total SOC contents is on average relatively small. In contrast, in certain situations earthworms can strongly affect greenhouse gas emission. These data were however mainly obtained in relatively short-term experiments. Over a period of months to years and even decades, earthworms can reduce C decomposition by physical protection of C in aging casts (Six et al., 2004, already cited in the MS).

Thus, long-lasting effects of invasive earthworms on the total SOC storage cannot be determined with certainty in short-term experiments, whereas field observations are rather controversial. For instance, (Wironen and Moore, 2006) reported ca. 30% increase in the total soil C storage in the earthworms-invaded sites of an old-growth beech-maple forest in Quebec. Other studies (e.g. (Sackett et al., 2013; Resner et al., 2014) suggest a decrease in C storage. (Zhang et al., 2013) introduced the sequestration quotient concept to predict the overall effect of earthworms on the C balance in soils of different richness, but the question remains strongly understudied.

These well documented examples of the impact of earthworms on soil C storage are related to invasive species. The presence of these species cannot be inferred directly from the climatic, soil and vegetation properties. The distribution of European invasive

earthworms in North America (and also in North European forests) is largely driven by human activity; often fishing (!) is more important than habitat transformation; without human's help earthworms are not active invaders (Stoscheck et al., 2012; Tiunov et al., 2006; Wironen and Moore, 2006). Thus the presence of earthworms is an "independent" parameter.

The term "humic substances" is now referred to in a more neutral way.

We also have elaborated in more detail why we addressed different biomes and land use types: not only because the key players generally differ, but also because their identity (e.g. which and how many species of earthworms or termites) and activity (modified abundance, burrowing activity etc.) may change with both climate and land use. In a global meta-analysis spanning several continents, (García-Palacios et al., 2013) show that across biomes and scales the presence of soil fauna contributes on average 27% to litter decomposition. Depending on the situation this contribution can be substantially lower or higher. For instance, the authors report an average increase in decomposition rates of 47% in humid grasslands whereas in coniferous forests this figure amounts to only 13%. The high impact of soil fauna in humid grasslands is all the more important as such grasslands are among those ecosystems that are most severely affected by global environmental change. (García-Palacios et al., 2013) also provide additional evidence on the argument that soil fauna activity is not merely a product of climate, soil properties and land use but an independent parameter. We offer some explanations for this phenomenon by referring to niche width (e.g. temperature or drought may become critical for both survival and invasion of species), and to the relevance of human activities for soil fauna beyond impact on global warming and land use change (e.g. pollution).

New references were added wherever appropriate. Any other referee comments were taken into account – if we did not do so we explained why in the text. Exceptions from this are: - Paper in review (recommended by referee 3) cannot be cited - Suggestions concerning the graphical quality will be taken care of when the final version is accepted

- The table of the manuscript does contain quite a number of figures detailing that treatments with fauna sometimes differ from the control without by even more than 40%. We made this more clear in the abstract

Acknowledgement We are confident that our manuscript has further improved and thank the referees for their thorough work. Sure we hope that the editor acknowledges our additional efforts.

References

García-Palacios, P., Maestre, F. T., Kattge, J. and Wall, D. H.: Climate and litter quality differently modulate the effects of soil fauna on litter decomposition across biomes, Ecol. Lett., 16(8), 1045–1053, doi:10.1111/ele.12137, 2013. Lubbers, I. M., van Groenigen, K. J., Fonte, S. J., Six, J., Brussaard, L. and van Groenigen, J. W.: Greenhouse-gas emissions from soils increased by earthworms, Nat. Clim. Chang., 3(3), 187–194, doi:10.1038/nclimate1692, 2013. Resner, K., Yoo, K., Sebestyen, S. D., Aufdenkampe, A., Hale, C., Lyttle, A. and Blum, A.: Invasive Earthworms Deplete Key Soil Inorganic Nutrients (Ca, Mg, K, and P) in a Northern Hardwood Forest, Ecosystems, 18(1), 89–102, doi:10.1007/s10021-014-9814-0, 2014. Sackett, T. E., Smith, S. M. and Basiliko, N.: Soil Biology & Biochemistry Indirect and direct effects of exotic earthworms on soil nutrient and carbon pools in North American temperate forests, Soil Biol. Biochem., 57, 459–467, 2013. Six, J., Bossuyt, H., Degryze, S. and Denef, K.: A history of research on the link between (micro)aggregates, soil biota, and soil organic matter dynamics, Soil Tillage Res., 79(1), 7–31, doi:10.1016/j.still.2004.03.008, 2004. Stoscheck, L. M., Sherman, R. E., Suarez, E. R. and Fahey, T. J.: Exotic earthworm distributions did not expand over a decade in a hardwood forest in New York state, Appl. Soil Ecol., 62, 124–130, doi:10.1016/j.apsoil.2012.07.002, 2012. Tiunov, A. V., Hale, C. M., Holdsworth, A. R. and Vsevolodova-Perel, T. S.: Invasion patterns of Lumbricidae into the previously earthworm-free areas of northeastern Europe and the western Great Lakes region of North America, Biol. Invasions Belowgr. Earthworms as Invasive Species, 23–34, doi:10.1007/978-1-4020-5429-7_4, 2006.

[Figure]

Wironen, M. and Moore, T. R.: Exotic earthworm invasion increases soil carbon and nitrogen in an old-growth forest in southern Quebec, Can. J. For. Res., 36(4), 845–854, doi:10.1139/x06-016, 2006. Zhang, W., Hendrix, P. F., Dame, L. E., Burke, R. a, Wu, J., Neher, D. a, Li, J., Shao, Y. and Fu, S.: Earthworms facilitate carbon sequestration through unequal amplification of carbon stabilization compared with mineralization., Nat. Commun., 4(OCTOBER 2013), 2576, doi:10.1038/ncomms3576, 2013.

Please also note the supplement to this comment:
http://www.soil-discuss.net/soil-2016-19/soil-2016-19-AC1-supplement.pdf
* * *
[Figure]

[Figure]

**Figure 3**: Flow scheme for an improved understanding of the role of soil fauna for soil organic matter (SOM) turnover. This scheme is basically followed within the COST Action ES 1406 (KEYSOM). Activities in A) and B) run parallel, followed by C) which ends with an improved SOM model. Exemplarily shown are scenarios for two biomes; the shaded miniature displays a different scale for one of them. Further explanations see text.

**Fig. 1.** Flow scheme for an improved understanding of the role of soil fauna for soil organic matter (SOM) turnover. This scheme is basically followed within the COST Action ES 1406 (KEYSOM). Activit

**Author comment – Soil Fauna: key to new carbon models**

**J. Filser et al.**

We appreciate the high quality and detail of the referees' comments, in particular by referee #1, which have righteously criticized several points that we improved in the revised version. Looking at the wealth of information asked for, it should be clear that it is impossible to cover all this within a single review article – especially with one written mainly by empiricists and pointing out research needs. In the new version we have better pronounced why we think that these needs are still there although the issue as such had been raised decades back: these are mainly a) missing information, b) too much detail, irrespective of spatial scale, and c) too little communication between empiricists and modellers.

Thus, and probably most importantly, we made now clear why we included the COST Action KEYSOM (which addresses right these issues and associated research needs), and added some more information on its contents and activities. Main aim of KEYSOM is to test the hypothesis that the inclusion of soil fauna activities into SOM models will result in a better mechanistic understanding of SOM turnover and in more precise process descriptions and predictions, at least locally. Current activities are mainly workshops and additional literature reviews, yet also modelling: Next to implementing fauna activity in already existing SOM models, a new basic SOM model structure is being developed. Ongoing literature reviews include faunal contribution to SOM turnover in different biomes and potentials and limitations of global SOM models for including soil fauna effects. Finally, a large-scale European field experiment into the impact of soil fauna composition and abundance on SOM breakdown, distribution and aggregate formation will start in autumn 2016.

Fig. 3 of the revised manuscript (see below) shows a form of a "roadmap" how KEYSOM is contributing to the challenges identified by our review. This is further explained in the accompanying text, where we also address a number of other issues raised by the referees. Please note that this roadmap mainly functions as a decision tree with potential actions and feedback loops at every step, for instance to initiate research (literature review or experimental studies) before proceeding further. Effects of fauna abundance or biomass (in comparison to presence-absence) on the shape of the function will be addressed as well (Fig. 3A, bottom). Note, however, that to date necessary data for such an approach appear to be limited (García-Palacios et al., 2013). As many existing models, also the new model should have a modular structure so that different modules can be used and combined according to the respective biome- and scale-specific scenario (Fig. 3C). It can also be seen that we do not aim to include every detail everywhere: in some situations (Fig. 3A) the impact of soil fauna on SOM dynamics might be too small (or existing information too scanty) to be included, and not all input parameters will be feasible or relevant at each scale (miniature in Fig. 3C). This keeps the model manageable, and also flexible enough to allow for more precise predictions in critical scenarios, like in the case of earthworm invasions sketched in the next paragraph. We generally think that focusing on such critical scenarios (analogous to e.g. global biodiversity hotspots) is a crucial precondition for well-informed management decisions, one of the final aims of KEYSOM.

To further substantiate our point of view, we followed the suggestion to add more quantitative figures. This was already partly shown in the table or in the text of the MS and further supplemented. In addition we included a more elaborate example based on the comparatively well studied impact of invasive earthworms. The meta-analysis of (Lubbers et al., 2013, already cited in the MS) suggests that the effect of earthworms on total SOC contents is on average relatively small. In contrast, in certain situations earthworms can strongly affect greenhouse gas emission. These data were however mainly obtained in relatively short-term experiments. Over a period of months to years and even decades, earthworms can reduce C decomposition by physical protection of C in aging casts (Six et al., 2004, already cited in the MS).

Thus, long-lasting effects of invasive earthworms on the total SOC storage cannot be determined with certainty in short-term experiments, whereas field observations are rather controversial. For instance,

**Fig. 2.** formatted author comment

---

## Author Response (AR1)

We appreciate the high quality and detail of the referees' comments, in particular by referee #1, which have righteously criticized several points that we improved in the revised version. Looking at the wealth of information asked for, it should be clear that it is impossible to cover all this within a single review article – especially with one written mainly by empiricists and pointing out research needs. In the new version we have better pronounced why we think that these needs are still there. Thus, and probably most importantly, we made now clear why we included the COST Action KEYSOM (which addresses right these issues and associated research needs), and added more information on its contents and activities. For details, please see subsequent point-by point replies. These are indicated in the marked-up manuscript by comments with the respective numbers.

| Referee comment | Point-by-point replies | Line |
|---|---|---|
| **Anonymous Referee #1** | | |
| The MS is intended to convince carbon modelers that they need to consider the effects of soil fauna in order to enhance the predictive power of the models and thereby develop more accurate accounting of soil carbon. But, here is where I think the MS falls short. While it presents important ideas, the material isn't communicated in a way that can be appreciated by modelers, who routinely tend not to be empiricists, let alone experimentalists, and therefore likely wouldn't concern themselves with the fascinating, but minutiae of, detailed mechanisms presented in the MS (or wouldn't know where to begin to incorporate the details into models). The MS instead would likely resonate most with soil biologists who are fascinated by the details, and thus the MS tends to "preach to the choir" so to speak. The MS would be strengthened if it played more directly to the perspectives or needs of modelers. But, here is where the authors need to decide on which direction to go. | 1.
We are very grateful to referee #1 and hope that the revised version will meet his or her main intention; see Fig. 3 and accompanying text. It is true that this paper does not give answers how to include fauna effects in the models. The aim of the paper was not to generate models but rather to show that by neglecting fauna modellers are missing an important variable. To do so we show that fauna plays an important role in many processes effecting SOM dynamics and that this fauna effect may not be already predicted by other parameters used in the models. | **540ff** |
| From where I stand, making a convincing case for including effects of soil animals in modeling needs to provide one of two important pieces of information (if not both). The first would be to provide modelers with a clearer sense of how discrepant their model predictions are because they don't include animal effects. That is, how much does the presence vs. absence of groups of soil animals influence the amount of soil carbon that is stored or lost. The MS gets at it a bit when discussing earthworm effects (line 344-346). But, there is scant other evidence provided to support the argument (beginning on line 347) that without considering the role of animals models will be less accurate. It would be helpful to know quantitatively how inaccurate the models will be if animal effects are not included (i.e., how much of a difference in carbon balance estimates is there?). I appreciate that this may be tough to do because empiricists and experimentalists aren't accustomed to examine soil biology and relate it to quantitative estimates of carbon balance in a form that is useable by modelers. (There is a lesson for empiricists here too). | 2.
This aspect falls beyond the scope of this article as we mainly aim for pointing out research needs and possible ways to proceed. However, it is being addressed within the current tasks of KEYSOM (COST ES 1406), which are now shown in more detail. | **130**
**500ff** |

| | | |
|---|---|---|
| Second, as the MS correctly points out, many of the biological details presented in the MS are ignored or simply subsumed as "parameters" in models that describe big-scale processes. This is typically done as a matter of mathematical convenience because abstracting a complex process as a parameter keeps the model reasonably tractable. But, accounting for animal effects in ways described in the MS requires characterizing those processes in terms of model functions, functions in which the levels of soil carbon due to a specific mechanism (e.g. soil bioturbation, or aggregate formation) vary with the abundance of the animals performing the mechanism. | 3.
See Fig 3 A (point 5) and accompanying text. Note that we also have taken into account the fact that often the available data will be insufficient.. | **566** |
| However, converting parameters into new functions can be a daunting exercise from two standpoints. First, empiricists and experimentalists tend to examine processes in terms of effects due to presence/absence of species and often do not vary animal abundance to measure what form a function should take. | 4.
See reply 3. We also pointed this out in the text. | **568** |
| Second, empiricists are enamored by biological details, but often don't give priority to which details might matter more than others. This can cause concern to modelers because including each and every detail can make the models vastly complex, therefore making model output extremely difficult to validate and therefore understand. So, given heterogeneity in soil properties across large geographic spaces, do we need to know accurately variation in local soil molecular structure, or local root processes, or local physical heterogeneity or local aggregate formation to inform regional carbon budgets? If yes, then how? What I am driving at is that the MS would be strengthened if it provided a better road map of what processes should be an immediate priority to include in modeling and what level of detail needs to be included in the models. | 5.
Our approach is not to include fauna effects in every carbon model anywhere but to take them into account in situations where the expected impact (as known from the literature) will be high. The decision tree in Figure 3 and the associated text detail this approach. | **540** |
| This road map could be strengthened if the MS could offer a sense of what the functional forms of the processes might look like (i.e., can we assume linearity? Must we consider nonlinearity? If nonlinearity, then what form should the nonlinear function take?). | 6.
Again, this is beyond the scope of this article, but KEYSOM is working on it (see replies no. 2 and 4), with scale playing a fundamental role for the functions. | **461** |
| Most importantly, if accounting for spatial variation in animal effects matters, then what is a reasonable spatial scale over which one can assume that animal effects are reasonably similar. That is, it would be impossible for models of regional carbon budgets to account for heterogeneity on a m^2 x m2 basis. What spatial scale could be reasonable: km^2 x km^2, 10 km^2 x 10 km^2? Solving this scaling problem is perhaps the most critical issue when trying to align models with empiricism. In my experience, this is what causes the biggest rift between modeling and empiricism: empiricists again tend to focus on details of very fine spatial heterogeneity and disagree with efforts to subsume that heterogeneity in to a reasonable large-scale spatial approximation. | 7.
We have taken into account the scaling problem, including the fact that often the type of relationship will vary with scale. See also replies no. 2 and 3. | **455**
**586** |

| | | |
|---|---|---|
| Ultimately, the issues raised in the MS are not merely issues that should be of concern just to modelers. Empiricists need to appreciate the challenges and demands of modeling and provide empirical input that can help meet the challenge by tailoring empirical estimates and analyses to explicitly inform modeling. There was a large movement afoot in ecology in the 1990's to do a better job of melding modeling and empirical work. Modern ecology seems not to have heeded that too much. Perhaps the important message of this MS is that we need to begin heeding this a lot more going forward. | 8.
Thank you very much! This is why KEYSOM came into being. We emphasized this argument. | **500ff** |
| | | |
| **Anonymous Referee #2** | | |
| At the current level of understanding soil biology processes, we do not need to justify the relevance of incorporating of soil fauna (or soil microorganisms, soil vertebrates etc.) into SOM models. | 9.
This might be true based on the known facts. However, these facts thus far have not "succeeded" for implementing the fauna into C models – unlike often very fine details on microbial community structure, for which there is definitely not enough data for model implementation. A very good example to illustrate this is the review by Schmidt et al. (2011) who did not mention the words "fauna" or "animal". Thus, we do feel the need to once more point out their relevance. See also Referee 3, point no. 18 below. | |
| Now the matter is to my knowledge how to implement these data in theoretical models. Why these incorporations were not done before: due to lack of communication between modellers and biologists? Or it was not done due to lack of technical support (software) in modelling? Or biologists were not "convincing" enough for theoretical modellers? | 10.
We believe in a combination of these issues, except for the technical issues/software. This is certainly the smallest problem, in particular with the region-specific modular approach suggested by us. See also reply no. 8. | **500ff** |
| Another matter in the ms is the introduction of the COST (KEYSOM) project? Is it like an advertisement of the project? What is the aim of this introduction? KEYSOM is a very interesting project, no doubts about that, but what is it for in the ms? Is it possible to justify the presence of such information in the review paper, please? | 11.
We agree that this had not been clear enough in the first version and improved this justification | **50ff** |
| I like how the authors are trying to classify the effects of soil fauna on soil processes and properties into so called "key insights". | 12.
Well, these had been introduced by Schmidt et al. – we just replied to these. | |
| It is nice to see that data are collected not only for earthworms effects but other soil invertebrates too, including Collembola, potworms, nematodes, ants, termites. Though, when looking at the source references in the Table 1 one gets an impression that there was not much done to quantitatively test effects of soil fauna in soil. It is the case, isn't it? | 13.
You are perfectly right as soil ecology thus far has mostly focused on nutrients, especially nitrogen, rather than on carbon turnover. Second, it was not our aim to extensively cover all literature. Rather, we pointed out some striking examples and provided a more detailed case on earthworms. Reviews of other fauna groups and specific biomes will partly be addressed within KEYSOM. | 530 |
| Concerning figures, I would like to add that after printing they both look fuzzy (not sharp). In Fig.1 in permafrost "star" the abbreviation PF is not seen at all. In general this Fig has more a look that I would describe as a "whatever" look. Soil microorganism symbol looks like something else. | 14.
The fuzziness is probably an effect of pdf conversion (the TIFF file is crystal clear). We did stick to the symbol character of this figure but redrew microorganisms and changed the colour of "PF" for better readability. | 1130 |

| | | |
|---|---|---|
| Is it possible for you to make a picture where clearly, for example, with arrows or spatial restructuring, will be seen which "insight" affects which animal-mediated process? | 15.
We refrained from doing this, for several reasons. First, our intention is not to show which "insights" affect what animal-mediated process but rather how animals affect these insights, e.g. creating soil depth by digging. Second, adding only the most important arrows would make the figure hopelessly confusing – one would need many more graphs for this purpose, and we did not want to blow up the manuscript too much | |
| In Fig.2 unit of MAP is mm per year. What is this dot after mm? | 16.
A mistake in the original graph. We removed it. Moreover, we coloured the figure, for consistency. | **1142** |
| Line 182. Needs to change references positions, year 2012 goes first.
Line 254. Two times is used the word "on", please change the sentence.
Line 290. Sentence ". . .diversity and abundance of soil animals IS reduced as. . ..", should not be ARE here instead?
Line 355. Needs to remove symbol % after 70. It should be only after 77%. | 17.
Done. | **220**
**223**
**298**
**342** |
| | | |
| **Anonymous Referee #3** | | |
| As pointed out by other reviewers, the paper contains a wealth of biological detail. However, I think that this detail leads to the main message of the paper (which I assume is the need for the SOM modelling community to incorporate fauna) being lost. | 18.
We hope that the revision, specifically Fig. 3 and text and the more detailed reasoning for including the COST Action, made up for the impression that the main message got lost.
Concerning the details, see reply no. 9 | **506ff** |
| To remedy this, I think the authors need to do four things:
a) If possible, clearly demonstrate how the presence or absence of fauna lead to changed model / empirical predictions of SOM dynamics. | 19.
Not the scope of this article, see reply no. 2. | |
| The controls for the examples in Table 1 are not always explained, so it is not clear how fauna are or are not important. | 20.
We now made this more explicit | **1077** |
| The modelling importance is touched upon around lines 347 but could be developed further, and explicit links could be provided in the earlier empirical review. | 21.
We did this at various points, except for the "explicit links", see reply no. 19 | **e.g.**
**414**
**450**
**461ff** |
| b) Clearly demonstrate that the already included model parameters (e.g. of climate, land use etc) do not adequately predict faunal composition (it is mentioned briefly with De Vries et al. 2013 on line 349). If land use/climate parameters can (generally) predict faunal composition then it is not immediately clear why fauna need including in SOM dynamic models if climate, land use etc can also predict SOM reasonably accurately [which I realise is not always the case]. | 22.
We extended the reference to De Vries somewhat and added one more concrete example for this. Otherwise please refer again to reply no. 19. | **421**
**450** |
| As noted by another review, do the discrepancies in model results mean that fauna need including, or is it that other processes (e.g. leaching, particulate loss, litter inputs) could need refining to improve model accuracy instead. | 23.
This is again what will be revealed by the modelling working group within KEYSOM and not the scope of this article. | **532** |

| | | |
|---|---|---|
| Acting on this comment though does depend on model aims - if the aim of the model is to provide prediction, do they need to be mechanistically accurate (in the extreme, can they actually be statistical?). If however the model is aimed at mechanistic understanding, then the need for faunal incorporation potentially becomes clearer. As noted by the authors, this aspect may become even more important when trying to account for environmental change. | 24.
This is now better specified, as we need improvement both in mechanistic understanding and predictions. The idea is to deal with this case-specifically, i.e. clear-cut experimental approaches and small scale for the mechanistic part and approaches based on keystone actors for better predictions at the regional scale. Including this should be optional, depending on the available information and expected impact (see also Fig. 3) | **540** |
| c) The authors mention in the Abstract that 'the contribution of soil fauna activities can be as high as 40 %' but it is not clear in the main text where this figure comes from. | 25.
We re-worded this. | **59** |
| More importantly, what is the distribution of faunal importance to SOM dynamics/stability in ecosystems? Is it that there is one study demonstrating this level of importance, but others only show a negligible contribution? If so, then perhaps the importance of incorporation of faunal activity into SOM models is being overblown. [I personally think we do need to think about its incorporation, but the evidence presented here is not as clear as it could be] It may be that with the data currently available the distribution cannot be assessed. If so, then this aspect should, in my opinion, at least be discussed. | 26.
As a matter of fact, this is known only for some examples. (By the same token, how much do soil scientists and modellers exactly know about the small-scale distribution of carbon or hydrological properties?). As a rule of thumb, fauna is positively related to carbon, simply for energetic reasons. We accordingly extended the sections related to biomes and the approaches within KEYSOM. | **359** |
| d) The title suggests 'new' models are required. It might be good if the authors could synthesize their review to put forward a conceptual model framework that contrasts with currently available frameworks e.g. RothC, CENTURY. | 27.
This is what we did – see Fig. 3 and related text. Contrasting our approach with existing models is the task of KEYSOM WG 2, see respective text | **540** |
| Abstract - line 60 - we suggest that inclusion of soil animal activities can fundamentally affect the predictive outcome of SOM models. This is a very strong statement which I do not think has been clearly demonstrated in the paper at present. Perhaps addressing a) to c) above will help justify this. I can only see one example in the paper (referring to earthworms in CENTURY) and am not sure this is a 'fundamental' difference in prediction. What is a fundamental difference in prediction anyway - differing magnitude, differing direction (e.g. carbon source or sink), something else...? | 28.
We kept this sentence, for the following reasons: First, we added yet another example for underlining our suggestion (!). Second, we did not write something like "we clearly demonstrated…" but did compile a lot of evidence on the quantitative importance of soil fauna for SOM turnover, including two concrete examples – justifying our "very strong statement".
A fundamental difference is anything substantial that quantitatively matters. Specifying this here is not sensible as it can be anything: e.g., if soil fauna increase soil C release only by 1% per year this will have a substantial impact on many processes after a decade or more. In turn, changing the direction of an element flow by 0.01% will not likely have any relevant impact even after decades. | |
| Line 68 - I don't think there is any need to advertise the COST action in the Abstract (it occupies 6 lines out of 24), and it would be better to finish with a strong statement of what this review paper has found and its implications. I would also remove reference to this COST action at the end of the main text - it seems like a weak ending to the paper unless this is the main message you want to communicate. | 29.
Oh yes, it is as KEYSOM is working on the implications! Still, we have now better pointed out this need– see also replies no.8, 11 and many other arguments above. | **71** |
| Line 104 - please provide more details on the correlative field study. Of what? What discrepancies? | 30.
This had already been detailed in the first version, | **442** |

| | | just in a different section – see lines 401ff | |
|---|---|---|---|
| Line 118 - point out regional differences in or of what? We already know about regional differences in climate, land use, soils? | 31. Done | | **136** |
| Line 151 - "modifications in molecular structure have significant effects on its [SOM] dynamics." This is presumably the important point in terms of justifying incorporation for modellers yet there are no references backing this statement up. In relation to point c) above, what is the quantitative importance? | 32. References have been added or were there before (Table 1). The quantitative importance of this is part of the modellers' task within KEYSOM. | | **1076 ff** |
| Line 155 - "the term humic substances is considered outdated" This is a very strong statement yet many modellers continue to use these conceptual pools which reflect different rates of organic matter turnover. You could alienate readers with such phrasing, likewise on line 328 with "As there is no scientific support for widespread belief in humic substances". | 33. We replaced this by more diplomatic wording. | | **186** |
| Indeed, current research is utilizing mid infrared spectroscopy to examine particulate, humic, and resistant organic carbon, operationally defined pools which match well with conceptual pools in the models. These results have then been validated against observed changes in soil C following land use change (Paul et al. 2016 in review; see also Baldock et al 2013 Soil Research 51: 577-595). Does this means SOM models need a fundamental overhaul or does it depend on what you are trying to predict? I don't think these statements are required for the broader message of the paper and would suggest tempering them. | 34. We are aware of this. Again, this is not a pedological paper. Our point is that these pools do depend on faunal activity – and we propose that predictions (not: measurements!) of such SOM pools might be improved based on known or modelled distributions of soil fauna. We revised the text; see respective sections, including references referring to humic substances in Table 1.. | | **166ff 186ff 421** |
| Line 180 - what happens to fire derived carbon in absence of soil fauna? | 35. This is outside the scope of our paper – especially as hardly anything is known on the process in the presence of fauna (yet; the field is emerging due to the interest in terra preta, biochar etc., and also from boreal sites (J. Bengtsson, pers. comm.) | | **234** |
| Line 220 - what does discussion of soil depth mean for SOM dynamics. Make the links explicit here and in the other sections. Line 241 - please put quantitative figures on 'a large contribution' i.e. define large. Line 273 - why does tensile strength matter for SOM dynamics. Line 289 - density of what? | 36. Done | | **259ff** |
| Line 311 - how does Figure 2 demonstrate that specific effects of soil organisms differ across space? How has increasing importance in humid-warm and nutrient-limited conditions been demonstrated? Does the absolute / relative difference with and without soil organisms increase in these conditions? Or is it that soil organic matter dynamics are faster in humid warm conditions and so the presence of animals is confounded with these climatic conditions? | 37. This figure is a compilation of textbook knowledge that has accumulated over many decades. Please refer to the original article from which it was taken (Brussaard et al. 2012). | | |
| Line 344 - this sentence could be clearer - was the slow C pool maintained when earthworms were present in the model? | 38. The sentence was complemented with the necessary information. | | **437** |
| Line 393 - "A number of workshops" [not 'workshop']. | 39. Done | | **515** |

**Title page**

**Soil fauna: key to new carbon models**
**Authors**

**Juliane Filser[1*], Jack H. Faber[2], Alexei V. Tiunov[3], Lijbert Brussaard[4], Jan Frouz[5],**
**Gerlinde De Deyn[4], Alexei V. Uvarov[3], Matty P. Berg[6], Patrick Lavelle[7], Michel Loreau[8],**
**Diana H. Wall[9], Pascal Querner[10], Herman Eijsackers[11], Juan José Jiménez[12]**

[1]Center for Environmental Research and Sustainable Technolgy, University of Bremen, General and Theoretical
Ecology, Leobener Str. – UFT, D-28359 Bremen, Germany.
email: filser@uni-bremen.de

* Corresponding author

[2]Alterra, Wageningen UR, Droevendaalsesteeg 3, 6708 PB Wageningen, The Netherlands
email: jack.faber@wur.nl

[3]Laboratory of Soil Zoology, Institute of Ecology & Evolution, Russian Academy of Sciences, Leninsky prospekt 33,
119071 Moscow, Russia
email: av.uvarov@hotmail.com
email: a_tiunov@mail.ru

[4]Dept. of Soil Quality, Wageningen University, P.O. Box 47, 6700 AA Wageningen, The Netherlands
email: lijbert.brussaard@wur.nl email: gerlinde.dedeyn@wur.nl

[5]Institute for Environmental Studies, Charles University in Prague, Faculty of Science, Benátská 2, 128 43 Praha 2,
Czech Republic
email: jan.frouz@natur.cuni.cz

[6]Vrije Universiteit Amsterdam, Department of Ecological Science, De Boelelaan 1085, 1081 HV Amsterdam, The
Netherlands
email: m.p.berg@vu.nl

[7]Université Pierre et Marie Curie, Centre IRD Ile de France, 32, rue H. Varagnat, 93143 Bondy Cedex, France
email: patrick.Lavelle@ird.fr

[8]Centre for Biodiversity Theory and Modelling, Station d'Ecologie Théorique et Expérimentale, UMR 5321 CNRS &
Université Paul Sabatier, 2, route du CNRS, 09200 Moulis, France
email: michel.loreau@ecoex-moulis.cnrs.fr

[9]School of Global Environmental Sustainability & Dept. Biology, Colorado State University, Fort Collins, CO
80523-1036, USA
email: Diana.Wall@ColoState.EDU

[10]University of Natural Resources and Life Sciences, Department of Integrated Biology and Biodiversity Research,
Institute of Zoology, Gregor-Mendel-Straße 33, A-1180 Vienna - Austria
email: pascal.querner@boku.ac.at

[11]Wageningen University and Research Centre, PO Box 9101, 6700 HB Wageningen, The Netherlands
email: Herman.Eijsackers@wur.nl

[12] ARAID, Soil Ecology Unit, Department of Biodiversity Conservation and Ecosystem Restoration, IPE-CSIC,
Avda. Llano de la Victoria s/n, Jaca 22700 (Huesca), Spain
email: jjimenez@ipe.csic.es

Please note that this is a marked-up version of the revised manuscripts. Numbers in the comments refer to our reply to the referee's comments.

[revised manuscript text omitted]

---

## Author Response (AR2)

| Anonymous Referee #3 | Comment no. | Line |
|---|---|---|
| Although the main text justifies the incorporation of the COST action, it is not so clearly demonstrated in the Abstract, and the way it is currently incorporated in the Abstract still reads like an 'advert' to me, while the last sentence re: field experiment, experimental data and so forth are too vague (in my opinion) for an Abstract. I believe the abstract would be stronger, if KEYSOM is still to be included, with a conclusion such as: "We argue that explicit consideration of the soil fauna is essential to make realistic modelling predictions on SOM dynamics and to detect expected non-linear responses to global change. We present a decision framework, to be further developed through the activities of KEYSOM, a European COST action, for when mechanistic SOM models should include soil fauna. The research activities of KEYSOM, such as field experiments and literature reviews, together with dialogue between empiricists and modellers, will inform how this is to be done." | 1. Thank you - we adopted this suggestion. | **68ff** |
| You could also include KEYSOM in the keywords if you want it to be searchable. | 2. Done | **77** |
| Key insights: Although some sections within this part now clearly demonstrate the link between the key insight and SOM dynamics (and then the importance of soil fauna in modifying the key insight) e.g. 2.6 on soil depth, other parts go straight into how animals affect the key insight, without stating how the key insight affects SOM. I realise the latter is included in Schmidt, but I think it might be clearer for readers to show how each insight affects SOM (briefly as with soil depth, not exhaustively), and then introduce the substantial body of the review regarding soil faunal effects on the key insight. | 3. Done | **137ff** |
| Table 1: I found the examples in Table 1 very interesting. However, I did not follow how many of the earthworm examples in 2. Humic substances, were to do with this insight, as they appeared most related to aggregate stability. It is also not immediately clear how e.g. casting 40-50 t/ha/yr influences 'physical heterogeneity' and so on and so forth. | 4. We added a few sentences to better explain this. | **296ff** |
| As another example, for 'Various or mixed groups' what does microbial grazing leading to leaching of DOC and $NH_4^+$ have to do with 'molecular structure'? Don't some of these substances already exist in the absence of animals, whereas your points, at least as I understood the main text, are to do with animals modifying the molecular structure of SOM and thus influencing its | 5. Please note that this always relates to the control without animals. Whenever leaching of DOC etc. is increased this means a transition of solid OM into the aquatic phase, subject to potential leaching, plant uptake or microbial metabolism. $NH_4^+$ is clearly an effect | **139ff** |

| | | |
|---|---|---|
| decomposition dynamics? | on molecular structure. Admittedly, the leaching of such an unspecific item like DOC cannot directly be related to an analogous process – yet on the other hand the assumption that increased leaching is merely an effect of desorption seems quite unlikely. Thus, we left this unchanged. | |
| In other words, for all of the examples, make it really clear how they relate to the key insights and the explanations in the main text. | 6. See answers no. 4 and 5 | |
| I am surprised there isn't at least some discussion as to whether other processes could be important in improving SOM models (e.g. dissolved and particulate organic matter leaching, soil erosion, better parameterisation of litter inputs), and not just the requirement to include animals. I think it would be good to briefly mention alternative explanations for current model discrepancies; I am not suggesting these are reviewed. | 7. We added this to the introduction | **88 ff** |
| I like the insertion of Figure 3. However, please explain what 'Etc' means. I would consider taking out the shaded miniature on the figure as I think it unnecessarily confuses and modify the main text accordingly. However, retain an explanation about what happens when data are not available at different scales. I think I found this somewhat confusing as to my geographical mind, 'biome' is a given scale e.g. boreal forest, tropical forest etc. Also, the flow from 6 to 7 to 8 is not clear, given the additional white box with 'compare output with previous models/versions'. I would consider redrawing this to a linear flow, similar to A) and B). | 8. Figure and text were modified accordingly, and the significance of "etc." in now correct spelling should be clear. | **581ff 947** |
| MINOR COMMENTS / GRAMMATICAL CORRECTIONS | | |
| I was surprised by the standard of the English in parts of the ms; I would suggest giving it a careful proof read before any final submission and, if accepted, publication. | 9. A native speaker has proof-read the manuscript. | **523** |
| Line 57: Suggest "Fauna control..." [fauna are plural, as are data later in the ms (line 423)], rather than "The fauna controls" | 10. Done (well, another referee once had corrected data to singular…) | **57** |
| Line 59: "We show lots of quantitative examples" sounds very colloquial; I would suggest deleting and just stating "We demonstrate a very strong impact of soil animals on carbon turnover" | 11. Done | **59** |
| Lines 88-89: key to understandING and predictING changes in global carbon cycling | 12. Done | **82-83** |
| Suggest "Some years ago" on line 96 be the start of a new paragraph. | 13. Done | **94** |
| Line 108: Suggest "A correlative large scale field study has shown that including soil animal activities could help clarify discrepancies in existing carbon models" | 14. Done (slightly modified) | **106f** |

| | | |
|---|---|---|
| Line 116: Suggest split this sentence to: "Here we use the 'key insights' proposed by ...on the relationship of soil fauna to SOM dynamics. Our review justifies the relevance of incorporating soil fauna into SOM models" | 15. Done | **115f** |
| When you state, on line 118, that there have been repeated reviews it begs the question why therefore you have done this review. Are SOM dynamics different to 'geological and pedological processes' - if so, I would state it explicitly, or just remove this sentence. | 16. Done | **118f** |
| Line 123 - the Dorn reference to ants seems far too detailed for an Introduction - I would place it within an appropriate part of the review of key insights. | 17. We moved it to Sect. 2.1 | **142ff** |
| Line 147/148 - I presume 'humification' means, in this context, 'decomposition dynamics of animal faeces'. If so, I would put the parentheses after animal faeces, not where it is presently. | 18. We corrected this to "humus" | **150** |
| It may also be worth clarifying what the relationship is (if there is one) between 'humification', 'humic substances' and 'humus' (both of which are introduced in the next section). Also, you define humic substances on line 371 in a more explicit way than in section 2.2 - I would insert the definition on line 371 into section 2.2. | 19. Done | **170ff** |
| Line 218: Suggest delete 'any'. | 20. Done | **234** |
| Line 294 - still unsure of what the relationship of tensile strength is to SOM dynamics - please elucidate. | 21. We added an explanation | **331ff** |
| Line 331-333: Reference(s) required for suggestion that humid grasslands are "among those ecosystems that are most severely affected by global environmental change" | 22. Two references were added. | **371f** |
| Line 341: "more detailed information on their biology is required". State why this is required for the modelling. As pointed out by review 1 (for the initial submission), it is very difficult for the modelling community to include lots and lots of details as it makes model interpretation and validation problematic. Your calls for more detailed biological information therefore needs to be justified from a modelling perspective. | 23. Supplemented accordingly | **380f** |
| Line 397: I still find the CENTURY example oddly phrased. You discuss 'implementing earthworm activity' and then state what happens 'without earthworms', all in the same sentence. Please rephrase. | 24. Done | **434ff** |
| Line 444: Unclear what 'richness' refers to. Could mean species richness, although could also mean e.g. soil fertility. I assume you mean fertility but not sure... | 25. Reworded | **482** |
| Line 472: THE main aim of KEYSOM...of soil fauna activity into SOM models... | 26. Done | **510** |
| Line 493 - provide a bit more detail about 'the development of a simple SOM model' - I presume you mean a model that explicitly incorporates faunal processes in it, but you don't actually state this. | 27. Done | **531** |

[revised manuscript text omitted]

**Figures**

[Figure]

**Figure 1**

[Figure]

**Figure 2**

[Figure]

[Figure]

[Figure]

**Figure 3**